# Counterfactual Visual Explanation via Causally-Guided Adversarial Steering

## Abstract

Recent work on counterfactual visual explanations has contributed to making artificial intelligence models more explainable by providing visual perturbations to flip the prediction. However, these approaches neglect the causal relationships and the spurious correlations behind the image generation process, which often leads to unintended alterations in the counterfactual images and renders the explanations with limited quality. To address this challenge, we introduce a novel framework CECAS, which leverages a causally-guided adversarial method to generate counterfactual explanations. It innovatively integrates a causal perspective to avoid unwanted perturbations on spurious factors in the counterfactuals. Extensive experiments demonstrate that our method outperforms existing state-of-the-art approaches across multiple benchmark datasets and ultimately achieves a balanced trade-off among various aspects of validity, sparsity, proximity, and realism.

## 1 Introduction

Despite the impressive performance of deep neural networks (DNNs) across domains such as language, vision, and control (Devlin et al., 2019; Bojarski et al., 2016; Silver et al., 2017), explaining their predictions remains challenging due to their non-linear, overparameterized nature (Li et al., 2022). This challenge is particularly critical in safety-sensitive applications like healthcare, finance, and autonomous driving (Mersha et al., 2024). As a result, explainable AI (XAI) has gained momentum (Dwivedi et al., 2023), with counterfactual explanations (CEs) Wachter et al. (2017) emerging as a powerful approach. CEs aim to answer: *What minimal change to an input would alter the model's prediction?* They are valued for producing intuitive and preferably actionable insights that can support decision auditing and planning. As a powerful XAI tool (Wachter et al., 2017), CE has been particularly flourishing in computer vision (Jacob et al., 2022; Jeanneret et al., 2022; Rodriguez et al., 2021; Jeanneret et al., 2023). Unlike attribution-based methods that highlight pixels associated with the output, counterfactuals in images simulate realistic alternative scenarios, allowing us to ask: What would the image need to look like for the model to decide differently? For example, modifying a few facial features to change a classification from "neutral" to "happy" provides insight into the model's decision boundary. Successful visual counterfactuals not only satisfy conventional CE criteria such as being *valid* (flip the prediction), *proximal* (close to the original image), and *sparse* (modify as few features as possible) (Mothilal et al., 2020; Russell, 2019), but also are semantically meaningful and perceptually natural, avoiding unrealistic artifacts. These methods offer intuitive and human-understandable explanations, making them especially valuable for analyzing high-dimensional visual models.

While counterfactual visual explanation has seen rapid progress in recent years Jacob et al. (2022); Jeanneret et al. (2022); Rodriguez et al. (2021); Jeanneret et al. (2023), several critical aspects of the problem remain overlooked. A key limitation lies in the lack of a principled understanding of the underlying causal mechanisms in the data, as well as the true target of explanation. Most existing methods (Petsiuk et al., 2018; 2021; Vasu & Long, 2020) optimize directly over the entire input feature space in search of perturbations, which often leads to changes in spurious factors that are correlated with the label but not causally related. While such perturbations may achieve a prediction flip, they rarely reflect meaningful or actionable changes in the real world and can even produce misleading or harmful outcomes. For example, in autonomous driving scenarios, changing the shape of a traffic sign (spurious factor) to alter a "stop" vs. "go" prediction is far less informative than changing the content of the sign, which is causally relevant to the model's decision.

More importantly, conventional CE strategies often result in low-quality and unrealistic images. The pixel space of images is vast, and unconstrained perturbations can introduce noise or artifacts that are visually implausible or semantically incoherent. Recent advances Jeanneret et al. (2022; 2023) have begun to address this by leveraging generative models, particularly diffusion-based methods, which operate in semantically meaningful latent spaces and are better equipped to produce realistic counterfactuals. These approaches benefit from the ability of diffusion models to suppress high-frequency noise and structure perturbations more coherently. However, most current methods still rely heavily on predefined loss functions, such as $l_1$ loss constraining the difference between original and counterfactual images, to enforce sparsity and proximity. While effective at a global level, these constraints often lead to unwanted changes in local artifacts such as color shifts, shape distortions, or the emergence of spurious attributes (Andonian et al., 2021). This results in a persistent trade-off between validity, proximity, sparsity, and realism, making it difficult to generate counterfactuals that are both high-quality and causally meaningful. To the best of our knowledge, no existing method well addresses these challenges, underscoring the need for a new approach that incorporates causal reasoning into the generation of counterfactual visual explanations.

To address the aforementioned challenges, we propose a novel framework, **C**ounterfactual **E**xplanation via **C**ausally **A**dversarial **S**teering (**CECAS**), which leverages a causality-guided adversarial method to generate counterfactual explanations that are more semantically faithful and causally grounded. Adversarial attacks have been an effective tool to generate CEs due to their common aim to minimally alter an input to change a model's prediction Pawelczyk et al. (2022). Uniquely, our approach integrates causality into the adversarial objective: While the adversarial component ensures prediction flips (validity), the causal guidance directs perturbations toward features that are causally relevant to the label. This promotes sparsity, proximity, and realism, by avoiding changes to spurious or irrelevant features. As a result, our method produces high-quality, semantically meaningful counterfactual images that align with real-world interventions. We conclude our contributions as follows:

- We provide a new perspective on the problem of counterfactual visual explanation by highlighting the critical role of causality. We show how causality can improve the quality of counterfactuals by avoiding spurious perturbations.

- We propose a novel framework with a causally guided adversarial approach to generate counterfactual visual explanations. Our method steers perturbations toward causally relevant features, enabling the generation of valid, sparse, proximal, and realistic counterfactual images.

- We conduct extensive experiments to evaluate our framework on multiple benchmark datasets. Our approach outperforms current state-of-the-art methods across various metrics both quantitatively and qualitatively.

## 2 RELATED WORK

**Explainable AI (XAI)** XAI refers to a set of techniques aimed at enhancing the transparency and explainability of AI models—making their decisions and internal mechanisms more understandable to humans (Mersha et al., 2024). The field is commonly divided into two major categories: ad-hoc and post-hoc methods. The key difference between them lies in the stage at which they are implemented (Guidotti et al., 2018). The former is often deployed during the training phase and advocates the design of inherently interpretable architectures (Alaniz et al., 2021; Alvarez Melis & Jaakkola, 2018; Bohle et al., 2021; Huang & Li, 2020; Nauta et al., 2021). In contrast, the latter focuses on understanding and explaining the trained models without modifying their internal structure (Selvaraju et al., 2017; Shrikumar et al., 2017; Simonyan et al., 2013). The post-hoc category includes global and local explanations, distinguished by the scope of the explanations they provide (Mersha et al., 2024). Global explanation provides a comprehensive description of the model (Lipton, 2018), while the latter focuses on explaining the model's prediction for a specific instance (Arrieta et al., 2020). The studied problem of our paper belongs to local explanations. This line of work mainly focuses on the following three directions: saliency map (Chattopadhay et al., 2018; Jalwana et al., 2021; Kim et al., 2022; Sobieski et al., 2024), concept attribution (Ghorbani et al., 2019; Kim et al., 2018; Kolek et al., 2022), and model distillation (Ge et al., 2021; Tan et al., 2018).

**Counterfactual Explanations** As a branch of post-hoc explanations, CE has emerged as a highly influential approach in recent years (Pawelczyk et al., 2022), especially in computer vision (Jeanneret et al., 2023). Some methods generate counterfactual samples by locating key regions that distinguish two contrasting images and substituting the relevant region from one image into the other (Goyal et al., 2019; Wang et al., 2021). Other methods utilize the gradient of the input image with respect to the target label while also leveraging robust models to generate human-recognizable modifications (Schut et al., 2021; Zhu et al., 2021). Additional approaches, including prototype-based algorithms (Van Looveren & Klaise, 1907) and invertible convolutional networks (Hvilshøj et al., 2021), have also shown promising abilities in generating counterfactual examples. Generative techniques are inherently well-suited for CE. With their rapid development, an increasing number of related works have emerged, which can be broadly categorized into conditional (Singla et al., 2019; Van Looveren et al., 2021) and unconditional approaches (Khorram & Fuxin, 2022; Rodriguez et al., 2021). Diffusion models (DDPM) are cutting-edge generative models. Jeanneret et al. (2022) and (Jeanneret et al., 2023) both utilize DDPM to generate counterfactual images. The difference lies in that the former modifies the sampling process of the DDPM, while the latter employs DDPM as a regularizer.

**Adversarial Attacks** Adversarial attacks and CEs share a similar objective: flipping the model's prediction, which has led some studies to focus on exploring the relationship between them (Akhtar et al., 2021; Ignatiev et al., 2019; Pawelczyk et al., 2022). However, the key difference lies in the fact that adversarial attacks on undefended models often result in imperceptible noise rather than semantically meaningful changes. Adversarial attacks include white-box and black-box settings. In white-box attacks, the attacker has access to the model's architecture and data (Costa et al., 2024). It typically generates adversarial examples by utilizing the gradient of the loss function with respect to the input (Goodfellow et al., 2014; Madry et al., 2017; Moosavi-Dezfooli et al., 2016). In contrast, black-box attacks (Andriushchenko et al., 2020; Wicker et al., 2018) restrict the attacker to accessing only samples from an oracle or pairs of inputs and corresponding outputs.

**Adversarial Defense** The development of adversarial attacks has, in turn, stimulated significant progress in adversarial defense. Recent works can be primarily divided into two categories: certified defense and empirical defense. Certified defense (Raghunathan et al., 2018; Cohen et al., 2019; Singla & Feizi, 2020) offer provable robustness by ensuring that the model's predictions remain stable under perturbations constrained within a specified norm ball. However, empirical defense (Goodfellow et al., 2014; Madry et al., 2017; Zhang et al., 2019), particularly adversarial training (Madry et al., 2017), is still highly effective. It typically generates adversarial samples by using an adversarial attack first and then incorporates such samples into the training process. It is worth noting that recent work has begun to integrate causality into adversarial defense strategies (Zhang et al., 2020; Carlini & Wagner, 2017; Mitrovic et al., 2020; Zhang et al., 2021), sharing a common objective with our work.

## 3 PRELIMINARIES

**Problem Definition** We first consider a classification task with $x$ representing an input image and $y$ standing for the corresponding label, where $x \in \mathcal{X}, y \in \mathcal{Y} := \{1, ..., K\}$, $\mathcal{X}$ and $\mathcal{Y}$ represent the image space and the label space, respectively. A DNN classifier trained for this task can be denoted by $f : \mathcal{X} \to \mathcal{Y}$. Then the counterfactual visual explanation can be defined as the task of designing a semantically meaningful, perceptible yet minimal perturbation $\delta$ that, when added to an input image $x$, generates a counterfactual image $x' := x + \delta$ and flips the classifier's prediction $\hat{Y}$ from the original prediction $y$ to a target label $y'$.

**Causal View on Image Generation** Figure 1 is a well-recognized causal graph $\mathcal{G}$ (Zhang et al., 2021) that can represent the generation processes of both the natural and counterfactual images. As shown, $C$ denotes the causal factors

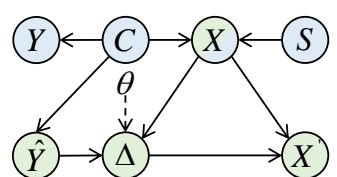

Figure 1: Causal graph of the natural and counterfactual images generation process. Each node means a variable, where blue nodes are involved in the generation of natural images, green nodes contribute to the generation of counterfactual images, and two-colored nodes participate in both.

that directly determine the label $Y$ and predicted label $\hat{Y}$. $S$ denotes the spurious factors, which are independent of $C$, as reflected in the $v$-structure $C \to X \leftarrow S$. $X$ is the natural image, and $\Delta$ represents the semantically meaningful perturbation; together, they form the counterfactual image

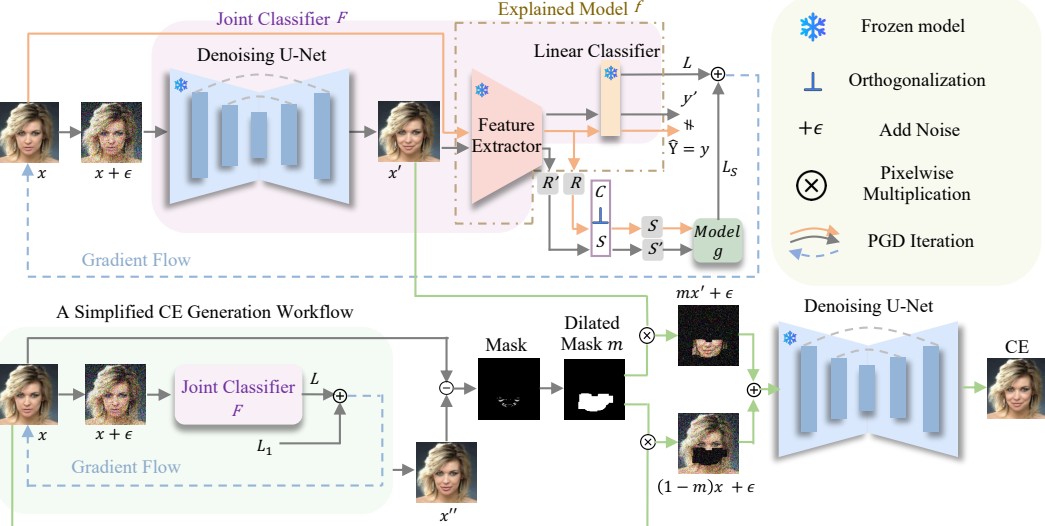

Figure 2: CECAS Framework: The upper part of the figure illustrates the first stage of the framework, which involves generating the initial CE images. Here, $x$ denotes the original image, $x'$ represents the causally-guided CE. The lower part of the figure depicts the second stage of the framework, namely post-refinement. Here, we refine $x'$ with a precise pixel-level mask $m$ (computed based on the difference between $x$ and a $l_1$-based CE $x''$), subsequently used in conjunction with image inpainting techniques to produce the high-quality final CE image. $R$ and $R'$ are the latent representations corresponding to $x$ and $x'$, respectively. $g$ is the auxiliary classifier.

$X'$. $\theta$ is the set of parameters of the DNN. It is evident from the graph that the perturbation $\Delta$ is determined by $\hat{Y}$, $X$ and $\theta$. The inclusion of $\theta$ is due to the use of white-box attacks, such as projected gradient descent (PGD). Note that $S$ can also have an influence on $Y$ because of the spurious paths in $\mathcal{G}$. When $X$ is given, there is a spurious path $Y \leftarrow C \rightarrow X \leftarrow S$ associate $Y$ and $S$.

**Denoising Diffusion Probabilistic Models** DDPM Ho et al. (2020) can filter out the noise introduced by traditional adversarial attacks without changing the classifier's weights. It relies on two Markov chains, corresponding to the forward and reverse processes, respectively. The forward process gradually adds Gaussian noise to the data at the state of $t$ to generate data at the next state $t + 1$. In practice, the data at time step $t$, denoted as $x_t$ can be directly obtained from the clean data $x_0$ through

$$x_t = \sqrt{\bar{\alpha}_t}x_0 + \sqrt{1 - \bar{\alpha}_t}\,\epsilon, \quad \epsilon \sim \mathcal{N}(0, I), \tag{1}$$

where $\bar{\alpha}_t := \prod_{s=1}^{t} \alpha_s$, $\alpha_t := 1 - \beta_t$ and $\{\beta_t\}_{t=1}^{T}$ is the variance schedule. The reverse process removes the noise from $x_t$ to get a less noisy $x_{t-1}$ through the following equation:

$$x_{t-1} = \frac{1}{\sqrt{\alpha_t}}\left(x_t - \frac{\beta_t}{\sqrt{1 - \bar{\alpha}_t}}\,\epsilon_\rho(x_t, t)\right) + \sigma_t\epsilon, \tag{2}$$

where $\sigma_t^2 = \beta_t$, $\epsilon_\rho$ is a function approximator intended to predict $\epsilon$ from $x_t$.

**Projected Gradient Descent Attack** PGD is a white-box attack that leverages the gradient of the loss function $L$ with respect to the input image $x$ to iteratively modify the image in a direction that maximally confuses the classifier. PGD operates through multiple iterations, and after each iteration $\tau$, the perturbed image $x_\tau$ is projected back onto an $l_\infty$-norm ball centered at the original input, with a predefined radius. This process can be summarized as follows:

$$x_{\tau+1} = \Pi_{x+\mathcal{S}}\left(x_\tau + \eta\,\mathrm{sgn}\left(\nabla_x L(\theta, f(x_\tau), y)\right)\right), \tag{3}$$

where $\Pi$ denotes the projection operator, $\mathcal{S}$ defines the radius of the $l_\infty$-norm ball, $\eta$ is the step size, sgn is the sign function, and $\theta$ represents the parameters of the target model.

## 4 METHODOLOGY

We propose the CECAS framework, which consists of two key components, as shown in Figure 2. The first component is the initial generation of CE images, achieved through a novel causally guided

adversarial method. The second component is the causally guided refinement of the generated counterfactuals, where we adopt an image inpainting approach to better preserve the details of the unmodified regions of the original image. We detail them in the following sections.

## 4.1 INITIAL CAUSALLY-GUIDED COUNTERFACTUAL VISUAL EXPLANATION GENERATION

The first and central component of the proposed framework is a causally guided adversarial method to generate counterfactual visual explanations. As previously mentioned, adversarial attacks and CE share a common goal: altering a model's prediction through minimal input perturbations. Inspired by prior work bridging the two domains (Pawelczyk et al., 2022; Jeanneret et al., 2022; Augustin et al., 2022), we adopt adversarial strategies—such as PGD attacks—to generate image-based counterfactuals. The procedure is outlined as follows:

$$x_{\tau+1} = \Pi_{x+\mathcal{S}} \left( x_\tau - \eta \operatorname{sgn} \left( \nabla_x L(f_\theta(x_\tau), y') \right) \right), \tag{4}$$

where the subscript $\tau$ denotes the iteration in PGD, $L$ is the classification loss, typically the cross-entropy loss, and $y'$ is the target label. Upon completing $T$ iterations, the resulting $x_T$ is taken as the counterfactual image we aim to generate. It is worth noting that, in form, Equation (4) differs from traditional PGD attacks. Here, the gradient is subtracted from the perturbed image rather than added. This is because the objective is not to mislead the classifier into making an arbitrary incorrect prediction, but rather to guide it toward a specific target label $y'$. This allows CEs to be generated with the gradient-based perturbation $\delta$ added on the original image.

However, relying solely on the above adversarial attack-based method is often overly idealized, as these methods typically impose no constraints on spurious factors. As discussed in the Introduction, although the explained model may rely on such factors during prediction, perturbing them often does not yield meaningful or actionable explanations. Furthermore, altering these spurious factors can introduce artifacts or noise, which severely compromise the core goals of CE of providing realistic and constructive explanations.

To address this issue, it is essential to constrain changes to spurious factors. Given the central role of gradients in the above CE generation process, we aim to identify a new gradient direction—one that enables effective modification of the image while preserving spurious factors unchanged. Since our goal is to explain the classifier $f$, we are not allowed to make any modifications to it. Considering the architecture of the explained model $f$, a DNN commonly viewed—following (Bengio et al., 2013)—as a combination of a feature extraction module and a linear classifier module, we learn to capture the spurious factors out of the representations $R$ learned by the feature extraction module (details of how to capture $S$ from $R$ are given in the final part of this subsection). With the captured $S$, we use the original data to train an auxiliary classifier $g$ which predicts the label $Y$ based on the spurious factor. More specifically, after training, $g$ can make predictions as follows: $\hat{z} = g(s), \hat{z}' = g(s')$, where $\hat{z}$ denotes the logit predicted by $g$ from the spurious factor $s$ in the original image, while $\hat{z}'$ is the logit predicted by $g$ from the spurious factor $s'$ in the modified image. We can consider $z$ to capture the spurious factors that are most relevant to the label $Y$—that is, the spurious features most likely to be unintentionally altered during counterfactual generation. On top of this, we introduce a penalty loss $L_S$ that encourages these spurious factors to remain invariant during the counterfactual generation process, and thereby we compute the gradient of the penalty loss with respect to the image and incorporate its direction into the guidance for image modification:

$$x_{\tau+1} = \Pi_{x+\mathcal{S}} \left( x_\tau - \eta \operatorname{sgn} \left( \nabla_x L(f_\theta(x_\tau), y') + \nabla_x L_S(z_\tau, z_\tau') \right) \right), \tag{5}$$

where $L_S$ can typically be implemented as the soft cross-entropy loss, which takes two logits as input. The purpose of this loss is to align the spurious distributions between the natural and counterfactual representations by penalizing the deviation of spurious factors in the counterfactuals from those in the original inputs. In summary, the generation of counterfactual images involves modifying the images via adversarial attack, while preserving the spurious factors through a causal perspective.

Furthermore, inspired by the effectiveness of diffusion models in enhancing CE image quality (Jeanneret et al., 2022; Augustin et al., 2022), we integrate diffusion models to filter out high-frequency, semantically meaningless noise introduced by adversarial attacks. Specifically, we apply the forward process of DDPM to add Gaussian noise to the image first, and then use the reverse process to denoise it before feeding it into the classifier for loss computation. In this work, we define the joint classifier $F$ as the combination of DDPM and the base classifier $f$. The generation of counterfactual images can be simplified and reformulated in an optimization form:

$$\arg\min_{x'} L(F(x'); y') + \alpha L_S(z, z'), \tag{6}$$

where $\alpha$ is the hyperparameter that controls the strength of $L_S$.

The only remaining problem is how to obtain the spurious factor $S$. Fortunately, numerous recent works have explored disentangling causal factors from spurious ones (Lv et al., 2022; Creager et al., 2021). Building on the causal model illustrated in Figure 1, we adopt the widely used assumption that the causal factor $C$ is independent of the spurious factor $S$. In this work, similar to (O'Shaughnessy et al., 2020), we pre-train a VAE $h$ to disentangle $C$ and $S$ from the latent space. $h$ takes the representation $R$ as input and is optimized with two objectives: (1) maximizing the causal influence of $C$ on $Y$; and (2) ensuring that $C$ and $S$, when jointly passed through the decoder, can faithfully reconstruct the distribution of $R$, thereby decomposing the latent space into two independent components.

## 4.2 CAUSALLY POST REFINEMENT

After generating the initial counterfactual images, a key challenge lies in ensuring **sparsity** and **proximity**. This is particularly difficult because the $L_S$ constraint operates at the concept level and often becomes "soft" when translated to pixel space, making it insufficient to enforce strict pixel-level consistency. While we have discussed the limitations of the $\ell_1$ loss, its advantages should not be overlooked—particularly its ability to identify precise pixel-level changes. Therefore, we leverage this property of $\ell_1$ loss to roughly localize pixels in the image that require modification, and then combine it with our causal constraint $L_S$ to avoid unnecessary changes to spurious factors. This supports our goal of *causally guided post-refinement*, where we preserve the counterfactual changes in $x'$ introduced by causal adversarial steering, while restoring the unmodified regions with the corresponding pixels from the original image $x$. Specifically, we first generate a perturbed image using an adversarial attack combined with $l_1$ loss as shown in the following equation:

$$\arg \min_{x''} L(F(x''); y') + \lambda ||x'' - x||_1, \tag{7}$$

where $x''$ is the perturbed image, $\lambda$ is a hyperparameter. Then a binary mask is constructed as:

$$m_{i,j} = \mathbb{1} \left[ \max_{(u,v) \in \mathcal{N}_{i,j}} \left( \frac{1}{M} \sum_{ch=1}^{CH} |x''(ch, u, v) - x(ch, u, v)| \right) < \gamma \right], \tag{8}$$

where $m_{i,j}$ is the binary mask value at position $(i, j)$, $x$ is the original image, $ch$ is the channel index, $(u, v)$ is the position index in an image, $CH$ is the number of channels, $M$ is the maximum value after the summation, $\mathcal{N}_{i,j}$ is a $d \times d$ neighborhood window centered at position $(i, j)$, $\mathbb{1}$ is the indicator function and $\gamma$ is a predefined threshold. We apply the dilation operation to adjust the size of the modification region to a reasonable extent, allowing for as many plausible changes as possible.

The mask indicates which pixels need to be modified through $l_1$ loss and which do not. Intuitively, we need to combine $x'$ generated in the last subsection and the original image $x$ with the mask to achieve causal perturbation in CEs with well-preserved pixels in the rest parts. However, directly combining $x'$ and $x$ using the mask is not feasible, as it would result in clearly visible region boundaries in the CE. In our work, we leverage the image inpainting technique (Lugmayr et al., 2022) to tackle this problem. The inpainting process is shown as follows:

$$x_{t-1} = \frac{1}{\sqrt{\alpha_t}} \left( [(1-m)x_t + mx_t^0] - \frac{\beta_t}{\sqrt{1-\bar{\alpha}_t}} \epsilon_\rho([(1-m)x_t + mx_t^0], t) \right) + \sigma_t \epsilon, \tag{9}$$

where $m$ is the binary mask, $x_t$ denotes the image being refined at the noise level $t$, and $x_t^0$ is the original image at the noise level $t$. After a pre-defined $T$ denoising steps, we can obtain the final counterfactual visual explanations $x'$ from the previous initial $x'$ we gained from Section 4.1.

## 5 EXPERIMENTS

In this section, we present extensive experiments to evaluate our proposed framework. We first compare our method against state-of-the-art baselines through both qualitative and quantitative analyses, including widely used metrics and case studies. Beyond traditional metrics, we further assess the quality of the generated counterfactuals using vision-language models (VLMs) as human-like evaluators. Additionally, we perform parameter studies to better understand the impact of different hyperparameter values in our method.

Table 1: Evaluation Results on CelebA-HQ: Smile.

| Methods | FR ↑ | COUT ↑ | LIPIPS ↓ | $S^3$ ↑ | CD ↓ | FID ↓ | sFID ↓ |
|---------|------|--------|----------|---------|------|-------|--------|
| DiVE | - | - | - | - | - | 107.0 | - |
| STEEX | - | - | - | - | - | 21.9 | - |
| DiME | 90.00% | -0.2797 | 0.1689 | 0.9245 | 2.52 | 15.8 | 26.79 |
| ACE | 77.50% | 0.2369 | 0.0353 | 0.9868 | 2.76 | 5.9 | 38.20 |
| **Ours** | **97.85%** | **0.6751** | **0.0331** | **0.9889** | **2.42** | **3.9** | **26.58** |

## 5.1 EXPERIMENTAL SETUP

**Datasets** We conducted extensive experiments on 4 datasets (10 subsets) in total. We first evaluate our framework on the CelebA dataset (Liu et al., 2015), using images of size 128×128 and focusing on two attributes which constitute three sub-datasets: "Smile", "Age" and "Gender". The CelebA-HQ (Lee et al., 2020) dataset is used with the same attributes and is also constructed into three sub-datasets, but with images of size 256×256. The other datasets include the traffic scene dataset BDD-OIA (Xu et al., 2020), with images of size 512×256, and three sub-datasets extracted from ImageNet (Russakovsky et al., 2015), each containing two classes. These three sub-datasets are: Cougar vs. Cheetah, Sorrel vs. Zebra, and Egyptian Cat vs. Persian Cat. All experiments were conducted on NVIDIA RTX 4090 and RTX A6000 GPUs.

**Baselines** We selected four recent and state-of-the-art methods as our baselines: STEEX (Jacob et al., 2022), DiVE (Rodriguez et al., 2021), DiME (Jeanneret et al., 2022), and ACE (Jeanneret et al., 2023). Among them, DiVE includes an extended variant called DiVE[100], in which the number of optimization steps is reduced to 100.

**Models Weights and Architectures** To ensure a fair comparison, we use the same pre-trained DDPM weights as those used in the baselines for each dataset. For ImageNet, we use the ResNet-50 model pre-trained by PyTorch (Paszke, 2019) as the classifier. For the other datasets, we use DenseNet-121 as the classifier and adopt the pre-trained weights provided by the baselines. Additional training details, including hyperparameter settings, are provided in Appendix A.

**Evaluation Metrics** We evaluate our method and the baselines across multiple dimensions, each assessed using several metrics. The first evaluation dimension is validity, which we measure using Flip Rate (**FR**) and Counterfactual Transition (**COUT**). FR denotes the proportion of samples for which the counterfactual successfully causes the classifier to change its prediction. COUT measures the transition scores between the original and counterfactual images, with values ranging from $-1$ to $1$. The second evaluation dimension focuses on sparsity and proximity, which we assess using Learned Perceptual Image Patch Similarity (**LPIPS**) (Zhang et al., 2018), Correlation Difference (**CD**) (Jeanneret et al., 2022), and SimSiam Similarity (**$S^3$**) (Chen & He, 2021). LPIPS measures the perceptual similarity between two images based on human visual perception. CD computes the average number of non-target attributes that are unintentionally altered across all samples. $S^3$ evaluates the cosine similarity between the original and counterfactual images using SimSiam as the encoding network. The third evaluation dimension is realism, which we measure using Fréchet Inception Distance (**FID**) (Heusel et al., 2017) and split FID (**sFID**) (Jeanneret et al., 2023). Since counterfactual images typically share most pixels with the original images, FID tends to be underestimated and may fail to reflect their true differences. To address this issue, sFID is proposed. It splits both the original and counterfactual image sets into two subsets, computes FID in a cross-wise manner, and repeats this process ten times to obtain an average score. All results reported in this paper are averaged over multiple experimental runs. We omit standard deviations as they are negligible compared to the corresponding mean values.

Table 2: Evaluation Results on CelebA-HQ: Age

| Methods | FR ↑ | COUT ↑ | LIPIPS ↓ | $S^3$ ↑ | CD ↓ | FID ↓ | sFID ↓ |
|---------|------|--------|----------|---------|------|-------|--------|
| DiVE | - | - | - | - | - | 107.5 | - |
| STEEX | - | - | - | - | - | 26.8 | - |
| DiME | 94.55% | -0.3339 | 0.1689 | 0.9202 | **4.00** | 17.8 | **26.54** |
| ACE | 63.45% | 0.0567 | **0.0448** | 0.9876 | 5.73 | 9.7 | 33.95 |
| **Ours** | **96.15%** | **0.5489** | 0.0466 | **0.9893** | 5.33 | **5.6** | 27.84 |

Table 3: Evaluation results on ImageNet.

| Datasets | Methods | FR ↑ | COUT ↑ | LPIPS ↓ | $S^3$ ↑ | FID ↓ | sFID ↓ |
|---|---|---|---|---|---|---|---|
| Sorrel-Zebra | ACE | 76.67% | 0.0967 | 0.1670 | 0.9285 | 77.76 | 124.66 |
| | **Ours** | **93.33%** | **0.2219** | **0.0468** | **0.9338** | **72.26** | **116.28** |
| Cougar-Cheetah | ACE | 76.67% | 0.2059 | 0.1809 | **0.9625** | 108.67 | 147.79 |
| | **Ours** | **100.00%** | **0.4186** | **0.0470** | 0.9616 | **102.56** | **137.91** |
| Persian Cat-Egyptian Cat | ACE | 80.00% | 0.2326 | 0.1531 | **0.8939** | 139.65 | 227.07 |
| | **Ours** | **83.33%** | **0.2660** | **0.0466** | 0.8744 | **134.89** | **215.07** |

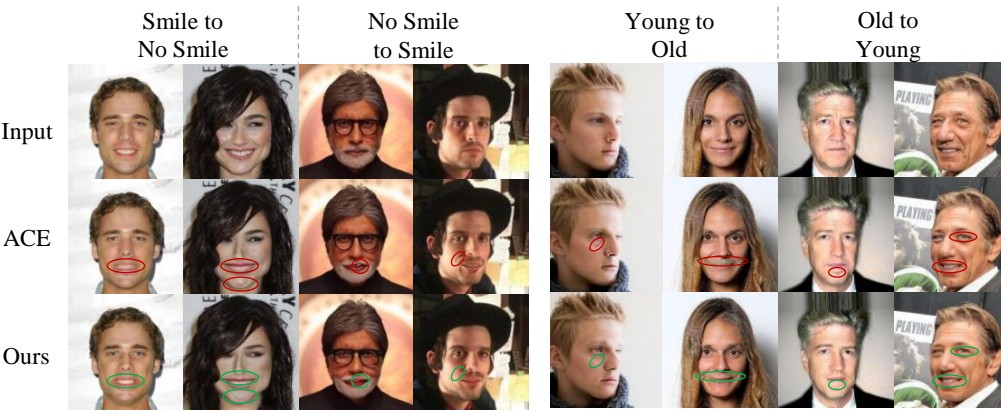

Figure 3: Qualitative results on CelebA (Smile & Age). Red circles highlight unintended changes produced by ACE, while green circles indicate the improvements made by our method.

## 5.2 QUANTITATIVE ANALYSIS

In this section, we conduct a quantitative analysis of all datasets with results summarized in Table 1 through Table 7. We highlight the best performance in **bold** and the second-best in underline. The results for the baselines DiVE, DiVE[100], and STEEX, we extract the results from the original papers. For the data not reported in the original papers, we leave the corresponding entries blank in the tables.

Tables 1and 2 show the results on the CelebA-HQ Smile and Age dataset, Table 3 shows the results on the ImageNet dataset. As observed, our method achieves the best performance on the vast majority of metrics. More specifically, our method achieves the highest FR and COUT scores, while also attaining the lower LPIPS and higher $S^3$ values, along with relatively low FID and sFID scores. This indicates that our method can achieve the highest flip rate with minimal modifications. In other words, it attains the highest validity while preserving both sparsity and proximity. Although in Table 2 our method performs slightly worse than DiME on certain metrics, the COUT of DiME is negative. This indicates that in DiME's method, the classifier lacks sufficient confidence in label flipping. Even though the other metrics may show marginal improvements, they do not necessarily indicate that the counterfactual images are of higher quality than those produced by our method. Due to page limitations, we present the results on the other datasets in Appendix B.

## 5.3 QUALITATIVE ANALYSIS

In this section, we perform case studies to straightforwardly evaluate the quality of the generated counterfactual images from a human perceptual perspective. For CelebA datasets, we compare the counterfactual images generated by our approach with those produced by the best-performing method, ACE, to demonstrate the visual superiority of our method. As shown in Figure 3, ACE introduces inexplicable color and shape distortions, as well as unintended attributes in the generated images. For example, in the transformation from "young" to "old," some female samples are incorrectly rendered with dark patches on the chin—an artifact caused by the spurious correlation between beard and age in male samples. This results in unmeaningful CEs. In contrast, our method effectively mitigates such artifacts by avoiding reliance on spurious cues. Figure 4 shows the results on ImageNet. As our method and ACE yield visually indistinguishable outputs on this dataset, we restrict the comparison to the original images and their counterfactual counterparts. Qualitative results for other datasets are provided in Appendix B.

## 5.4 VLM Evaluations

Vision-Language Models (VLMs), through large-scale pretraining on image–text pairs, demonstrate strong reasoning capabilities in tasks such as visual question answering and image captioning (Bordes et al., 2024). Overcoming the limitation of traditional CE metrics that only provide shallow numerical evaluations and fail to capture the semantic meaning of images, VLMs provide more human-like evaluations by leveraging their rich prior knowledge in pertaining (Huang et al., 2025). To better suit the specific nature of counterfactual image

Table 4: Performance comparison across datasets.

| Datasets | DiME | ACE | **Ours** |
|---|---|---|---|
| CelebA Smile | 7.49 | 8.14 | **8.24** |
| CelebA Age | 7.06 | 7.13 | **7.57** |
| CelebA-HQ Smile | 6.18 | 8.19 | **8.55** |
| CelebA-HQ Age | 3.95 | 5.76 | **6.19** |
| BDD-OIA | 4.07 | **7.86** | 7.77 |
| Sorrel-Zebra | - | 5.73 | **6.37** |
| Cougar-Cheetah | - | 7.14 | **7.61** |
| Persian Cat-Egyptian Cat | - | 6.93 | **7.63** |

generation, we carefully design unique prompts with evaluation criteria covering validity, sparsity, proximity, and realism. The prompt design for each dataset can be found in Appendix C. Notably, sparsity in our prompt is partially assessed from a causal perspective, examining whether the modified regions in the image are necessary or causally relevant, while ensuring that unrelated parts remain unchanged. In the context of our task, we utilize GPT-4o to assign scores ranging from 1 to 10 for each of the four evaluation dimensions. Given that these dimensions hold equal importance in counterfactual generation, the final score is computed as the average across all four. As shown in Table 4, our method outperforms the baselines in 7 out of the 8 sub-datasets. Combined with the powerful visual and textual reasoning capabilities of VLMs, we have strong reason to believe that the counterfactual images generated by our method offer the highest level of interpretability.

## 5.5 Parameters Study

In this section, we analyze the impact of the hyperparameter $\alpha$ on counterfactual image generation. We present results on the CelebA: Smile dataset, while results on the other datasets are provided in Appendix D. As shown in Figure 4, when $\alpha$ increases, the model is progressively constrained from modifying spurious factors. This trend leads to a monotonic decrease in both the flip rate and the COUT score. The reduction in flip rate indicates that many successful counterfactual transitions at lower

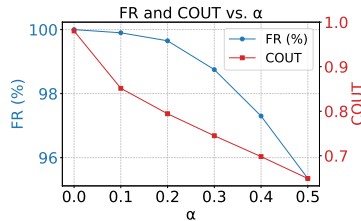

Figure 4: Line plots of FR and COUT with respect to varying values of $\alpha$.

$\alpha$ values are largely driven by spurious shortcuts; once these features are restricted, generating effective counterfactuals becomes more challenging. The concurrent decline in COUT further suggests that the transition trajectories become less smooth, with the classifier maintaining high confidence in the original class and showing limited confidence shifts toward the target class throughout the interpolation path. Taken together, these results demonstrate that higher $\alpha$ enforces a stricter focus on causal features, thereby producing counterfactuals that are harder to realize but more faithful to causal semantics. Therefore, it is crucial to strike a balance between flip effectiveness and visual realism. Fortunately, with a properly chosen $\alpha$, our method effectively achieves this balance, producing counterfactuals with both high flip rates and high image quality.

## 6 Conclusion

In this paper, we propose a novel framework, CECAS, which innovatively designs a causally-guided adversarial method to generate counterfactual visual explanations that constrain less informative changes on spurious factors. The framework can be applied to any DNN models. Experimental results demonstrate that our method consistently outperforms existing approaches in quantitative evaluation, qualitative analysis, and VLM-based assessment. CECAS achieves a better balance among the four key properties of counterfactual explanations: validity, sparsity, proximity, and realism. Therefore, we argue that the counterfactual explanations generated by our method offer a more causally grounded understanding of the classifier's decision-making and prediction mechanisms.

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

# Appendix

## A  TRAINING DETAILS

### A.1  DATA PREPARATION

The ten datasets discussed in the main text are categorized into two types: binary datasets and multi-label datasets. CelebA Smile, CelebA Age, CelebA Gender, CelebA-HQ Smile, CelebA-HQ Age, CelebA-HQ Gender, and BDD-OIA belong to the first category, while Cougar vs. Cheetah, Sorrel vs. Zebra, and Egyptian Cat vs. Persian Cat fall into the latter. For binary-label datasets, if the original predicted label is $y$ ($y$ can either be 0 or 1), then the target label is set to $1 - y$. For multi-label datasets, since ImageNet contains 1,000 classes, the prediction and target labels differ for each dataset subset. Specifically, Cougar vs. Cheetah: predicted label is 286, corresponding to "cougar"; target label is 293, corresponding to "cheetah". We adopt this setting to perform a transition from the "cougar" class to the "cheetah" class. Sorrel vs. Zebra: The predicted label is 339, which corresponds to "sorrel"; the target label is 340, which corresponds to "zebra". We adopt this setting to perform a transition from the "sorrel" class to the "zebra" class. Egyptian Cat vs. Persian Cat: predicted label is 283, corresponding to "Persian cat"; target label is 285, corresponding to "Egyptian cat". We adopt this setting to perform a transition from the "Persian cat" class to the "Egyptian cat" class.

### A.2  HYPERPARAMETERS SETTINGS

We provide the detailed hyperparameter settings for each dataset in Table 5. $\alpha$ is the weight of the adversarial defense loss, $\lambda$ is the weight of the $l_1$ loss used during mask generation, and $\gamma$ is the threshold for generating the binary mask. For the VAE architecture, we use a linear encoder and decoder across all datasets. The encoder consists of a linear layer followed by a ReLU activation. The input dimension is 1024, except for ImageNet, where it is 2048. The output dimension is 512. This is followed by two separate linear layers to compute the mean and variance, each with an input dimension of 512 and an output dimension of $d_C + d_S$ ($d_C$ is the dimension of causal representation; $d_S$ is the dimension of spurious representation), respectively (see Table 6 for specific values). The decoder architecture mirrors that of the encoder.

Table 5: Hyperparameter Settings for All Datasets.

| Datasets | $\alpha$ | $\lambda$ | $\gamma$ | PGD Iterations | Diffusion Steps |
|---|---|---|---|---|---|
| CelebA Smile | 0.2 | 0.001 | 0.1 | 50 | 500 |
| CelebA Age | 0.5 | 0.001 | 0.1 | 50 | 500 |
| CelebA Gender | 0.4 | 0.001 | 0.1 | 50 | 500 |
| CelebA-HQ Smile | 0.01 | 0.001 | 0.1 | 50 | 500 |
| CelebA-HQ Age | 0.03 | 0.001 | 0.1 | 50 | 500 |
| CelebA-HQ Gender | 0.005 | 0.001 | 0.1 | 50 | 500 |
| BDD-OIA | 0.005 | 0.001 | 0.05 | 50 | 1000 |
| Sorrel-Zebra | 0.03 | 0.001 | 0.15 | 100 | 1000 |
| Cougar-Cheetah | 0.02 | 0.001 | 0.15 | 100 | 1000 |
| Persian Cat-Egyptian Cat | 0.04 | 0.001 | 0.15 | 100 | 1000 |

## B  ADDITIONAL EXPERIMENTAL RESULTS

Due to the page limit, we omit the results of CelebA, CelebA-HQ Gender, and BDD-OIA from the main text and provide them in this section as supplementary material. We first present the quantitative analysis. Table 7, Table 8, and Table 9 summarize the results on the CelebA dataset, Table 10 concludes the results on CelebA-HQ Gender, which exhibit a similar overall trend to that observed on CelebA-HQ Smile and Age. Although ACE achieves slightly better performance than our method on LPIPS and $S^3$, this comes at the cost of reduced flip rate and FID. Importantly, FID is a key metric, as it measures whether the overall distribution of counterfactual images is close to that of real images. Similarly, although DiME attains slightly better results on sFID, its COUT is extremely low and even negative, indicating that the classifier lacks confidence in the counterfactual

Table 6: VAE Hyperparameter Settings for All Datasets.

| Datasets | $d_C$ | $d_S$ | Training Iteration |
|---|---|---|---|
| CelebA Smile | 8 | 56 | 1000 |
| CelebA Age | 8 | 56 | 1000 |
| CelebA Gender | 8 | 56 | 1000 |
| CelebA-HQ Smile | 16 | 112 | 5000 |
| CelebA-HQ Age | 16 | 112 | 5000 |
| CelebA-HQ Gender | 16 | 112 | 5000 |
| BDD-OIA | 16 | 112 | 5000 |
| Sorrel-Zebra | 32 | 224 | 5000 |
| Cougar-Cheetah | 32 | 224 | 5000 |
| Persian Cat-Egyptian Cat | 32 | 224 | 5000 |

Table 7: Evaluation Results on CelebA: Smile.

| Methods | FR ↑ | COUT ↑ | LPIPS ↓ | $S^3$ ↑ | CD ↓ | FID ↓ | sFID ↓ |
|---|---|---|---|---|---|---|---|
| DiVE | - | - | - | - | - | 29.4 | - |
| DiVE$^{100}$ | - | - | - | - | 2.34 | 36.8 | - |
| STEEX | - | - | - | - | - | 10.2 | - |
| DiME | 97.60% | -0.0451 | 0.0896 | 0.9158 | 1.99 | 12.3 | 25.48 |
| ACE | 99.05% | 0.6547 | 0.0273 | 0.9822 | 2.05 | 4.2 | 26.38 |
| **Ours** | **99.65%** | **0.7943** | 0.0299 | 0.9819 | 2.29 | **3.9** | 25.70 |

transitions produced by DiME. Table 11 presents the results on the BDD-OIA dataset. Although ACE performs well on several metrics, our method outperforms it in terms of COUT, demonstrating that the classifier retains sufficient confidence in the counterfactual transitions.

We then present the qualitative analysis. Since ACE achieves the strongest performance among all baselines, we focus our comparison on the counterfactual images generated by our method and those produced by ACE. As shown in Figure 5, similar to the results on the CelebA dataset, the counterfactual images generated by ACE contain many unintended modifications. These include changes in teeth and chin color, the appearance of exaggerated facial creases, and the emergence of extraneous patches on the cheeks. In the transition from "old" to "young", the counterfactual images generated by our method exhibit comparable visual quality to those produced by ACE, with no obvious undesired changes in either case. As a result, we do not highlight any specific regions for this comparison. However, our method aligns more closely with the intended transformation—for example, by effectively removing age spots and smoothing wrinkles—demonstrating greater semantic consistency with the target concept. In the transition from "male" to "female", we omit the highlight for the same reason.

The extended qualitative results on CelebA, CelebA-HQ, BDD-OIA, and ImageNet are presented in Figures 5 to Figure 10. On CelebA and CelebA-HQ, we highlight the unintended alterations present in the counterfactual images generated by ACE, while also marking the corresponding regions where our method yields noticeable quality improvements. For BDD-OIA and ImageNet, since the visual differences between our method and the baseline are minor, we only present the counterfactual images generated by our method, with the regions of change relative to the original images highlighted.

## C  VLM PROMPT DESIGN

In this section, we provide the prompt designs for each dataset to facilitate evaluation using Vision-Language Models (VLMs). While the evaluation dimensions are consistent with traditional metrics,

Table 8: Evaluation Results on CelebA: Age

| Methods | FR ↑ | COUT ↑ | LPIPS ↓ | $S^3$ ↑ | CD ↓ | FID ↓ | sFID ↓ |
|---|---|---|---|---|---|---|---|
| DiVE | - | - | - | - | - | 33.8 | - |
| DiVE$^{100}$ | - | - | - | - | - | 39.9 | - |
| STEEX | - | - | - | - | - | 11.8 | - |
| DiME | 73.10% | 0.0381 | 0.0938 | 0.9158 | 3.81 | 20.2 | 34.37 |
| ACE | 99.00% | 0.5441 | 0.0326 | 0.9843 | 3.37 | 4.3 | 26.47 |
| **Ours** | **99.75%** | **0.6760** | 0.0333 | 0.9829 | **3.17** | 4.3 | 25.76 |

Table 9: Evaluation results on CelebA: Gender.

| Methods | FR ↑ | COUT ↑ | LPIPS ↓ | S³ ↑ | CD ↓ | FID ↓ | sFID ↓ |
|---------|------|--------|---------|------|------|-------|--------|
| DiME | 97.05% | 0.0636 | 0.0914 | 0.9158 | 4.91 | 14.3 | 27.9 |
| ACE | 91.75% | 0.4947 | 0.0360 | 0.9826 | 5.44 | 7.3 | 29.6 |
| **Ours** | **97.15%** | **0.6304** | 0.0387 | 0.9807 | **4.53** | **6.3** | **27.9** |

Table 10: Evaluation results on CelebA-HQ: Gender.

| Methods | FR ↑ | COUT ↑ | LPIPS ↓ | S³ ↑ | CD ↓ | FID ↓ | sFID ↓ |
|---------|------|--------|---------|------|------|-------|--------|
| DiME | 98.45% | -0.3630 | 0.1671 | 0.9146 | 7.55 | 17.3 | 28.0 |
| ACE | 26.75% | -0.3705 | 0.0376 | 0.9865 | 9.66 | 26.7 | 55.4 |
| **Ours** | **99.95%** | **0.8555** | 0.0627 | 0.9831 | 8.30 | **7.6** | 29.0 |

the key distinction lies in our use of a VLM as a querying agent. Rather than relying solely on conventional metrics—which may be limited in capturing perceptual or semantic nuances—we engage the VLM through human-centric, dialog-style questions. This enables more holistic and perceptually grounded evaluations, aligning better with human judgment and interpretability.

**CelebA Smile** *You are given two sets of facial images: one set contains the original images, and the other contains the corresponding counterfactual images. The target attribute to be modified is smile. If the person in the original image is smiling, the target is "not smiling"; if the person is not smiling, the target is "smiling". Please evaluate the counterfactual images based on the following four criteria, each rated on a scale from 1 to 10:*

- *Validity: Does the counterfactual image successfully transition to the target attribute (i.e., smiling or not smiling) in a visually convincing way?*
- *Sparsity: Are irrelevant attributes and regions left unchanged, with only the necessary parts being modified?*
- *Proximity: From a human perceptual perspective, how close is the counterfactual image to the original one?*
- *Realism: Does the counterfactual image remain close to the data manifold and distribution, i.e., does it appear as a realistic image that could exist in the original dataset?*

**CelebA Age** *You are given two sets of facial images: one set contains the original images, and the other contains the corresponding counterfactual images. The target attribute to be modified is age. If the person in the original image is old, the target is "young"; if the person is young, the target is "old". Please evaluate the counterfactual images based on the following four criteria, each rated on a scale from 1 to 10:*

- *Validity: Does the counterfactual image successfully transition to the target attribute (i.e., young or old) in a visually convincing way?*
- *Sparsity: Are irrelevant attributes and regions left unchanged, with only the necessary parts being modified?*
- *Proximity: From a human perceptual perspective, how close is the counterfactual image to the original one?*
- *Realism: Does the counterfactual image remain close to the data manifold and distribution, i.e., does it appear as a realistic image that could exist in the original dataset?*

**CelebA Gender** *You are given two sets of facial images: one set contains the original images, and the other contains the corresponding counterfactual images. The target attribute to be modified is age. If the person in the original image is male, the target is "female"; if the person is female, the target is "male". Please evaluate the counterfactual images based on the following four criteria, each rated on a scale from 1 to 10:*

- *Validity: Does the counterfactual image successfully transition to the target attribute (i.e., male or female) in a visually convincing way?*
- *Sparsity: Are irrelevant attributes and regions left unchanged, with only the necessary parts being modified?*

Table 11: Evaluation results on BDD-OIA.

| Methods | FR ↑ | COUT ↑ | LPIPS ↓ | $S^3$ ↑ | FID ↓ | sFID ↓ |
|---|---|---|---|---|---|---|
| DiME | 60.50% | -0.4865 | 0.1119 | 0.9638 | 62.3 | 225.17 |
| ACE | 99.90% | 0.7278 | 0.0318 | 0.9956 | 9.5 | 91.70 |
| **Ours** | 99.50% | **0.7745** | 0.0449 | 0.9949 | 15.3 | 92.35 |

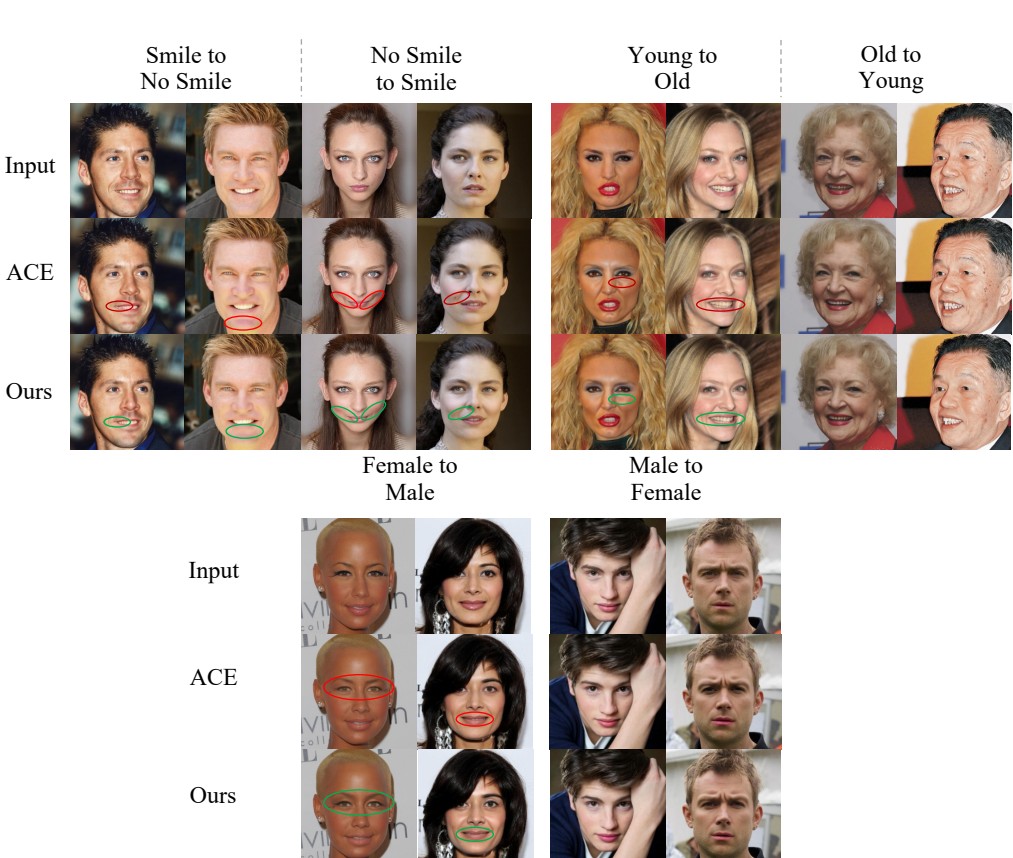

Figure 5: Qualitative results on CelebA-HQ. Red circles highlight unintended changes produced by ACE, while green circles indicate the corresponding improvements made by our method.

- *Proximity: From a human perceptual perspective, how close is the counterfactual image to the original one?*

- *Realism: Does the counterfactual image remain close to the data manifold and distribution, i.e., does it appear as a realistic image that could exist in the original dataset?*

**CelebA-HQ Smile** *You are given two sets of facial images: one set contains the original images, and the other contains the corresponding counterfactual images. The target attribute to be modified is smile. If the person in the original image is smiling, the target is "not smiling"; if the person is not smiling, the target is "smiling". Please evaluate the counterfactual images based on the following four criteria, each rated on a scale from 1 to 10:*

- *Validity: Does the counterfactual image successfully transition to the target attribute (i.e., smiling or not smiling) in a visually convincing way?*

- *Sparsity: Are irrelevant attributes and regions left unchanged, with only the necessary parts being modified?*

- *Proximity: From a human perceptual perspective, how close is the counterfactual image to the original one?*

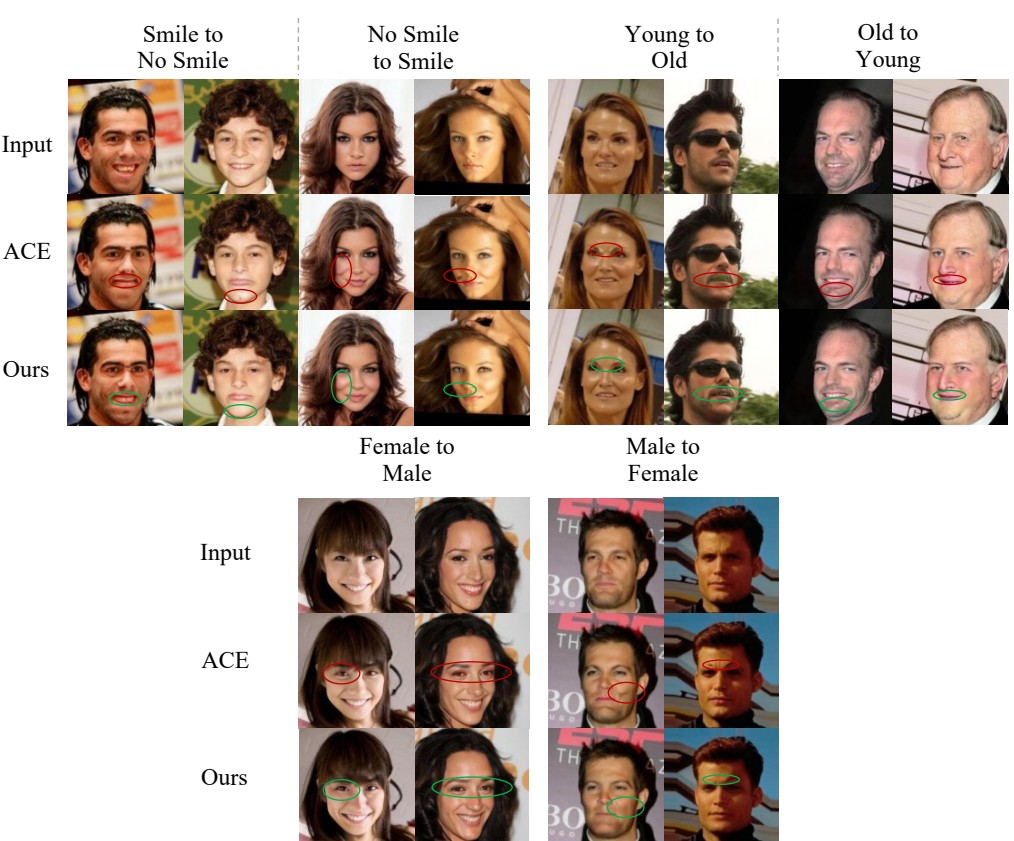

Figure 6: Extended qualitative results on CelebA. Red circles highlight unintended changes produced by ACE, while green circles indicate the corresponding improvements made by our method.

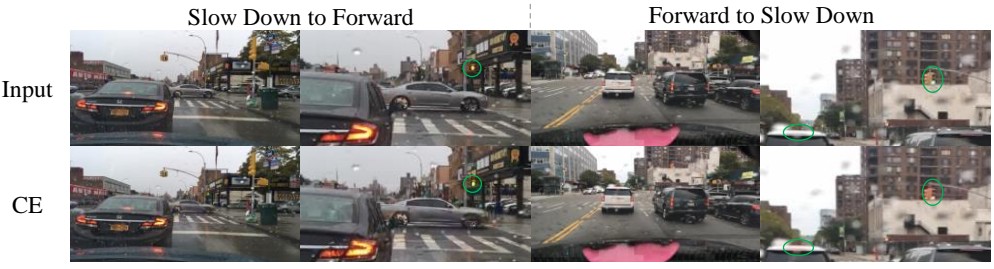

Figure 7: Qualitative results on BDD-OIA. Green circles highlight the counterfactual changes. The right column for each transition direction shows a zoomed-in version.

- *Realism: Does the counterfactual image remain close to the data manifold and distribution, i.e., does it appear as a realistic image that could exist in the original dataset?*

**CelebA-HQ Age** *You are given two sets of facial images: one set contains the original images, and the other contains the corresponding counterfactual images. The target attribute to be modified is age. If the person in the original image is old, the target is "young"; if the person is young, the target is "old". Please evaluate the counterfactual images based on the following four criteria, each rated on a scale from 1 to 10:*

- *Validity: Does the counterfactual image successfully transition to the target attribute (i.e., young or old) in a visually convincing way?*

**Slow Down to Forward**          **Forward to Slow Down**

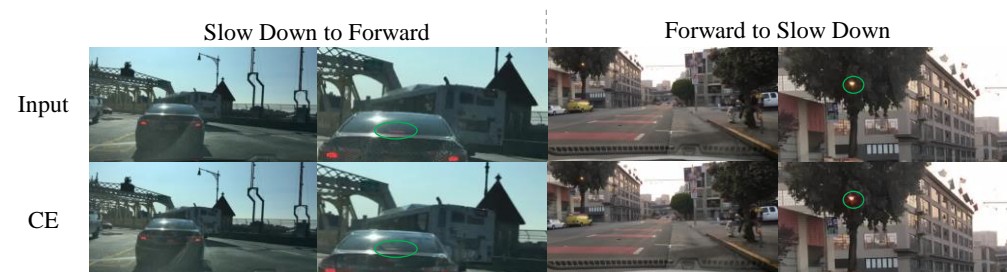

Figure 8: Extended qualitative results on BDD-OIA. Green circles highlight the counterfactual changes. The right column for each transition direction shows a zoomed-in version.

**Sorrel to Zebra**          **Cougar to Cheetah**          **Persian Cat to Egyptian Cat**

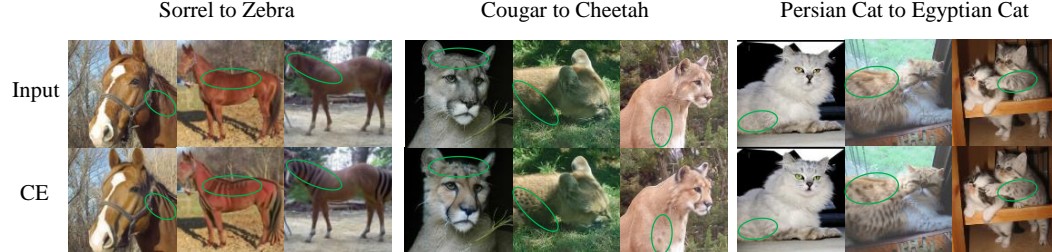

Figure 9: Qualitative results on ImageNet. Green circles highlight the counterfactual changes.

- *Sparsity: Are irrelevant attributes and regions left unchanged, with only the necessary parts being modified?*

- *Proximity: From a human perceptual perspective, how close is the counterfactual image to the original one?*

- *Realism: Does the counterfactual image remain close to the data manifold and distribution, i.e., does it appear as a realistic image that could exist in the original dataset?*

**CelebA-HQ Gender** *You are given two sets of facial images: one set contains the original images, and the other contains the corresponding counterfactual images. The target attribute to be modified is age. If the person in the original image is male, the target is "female"; if the person is female, the target is "male". Please evaluate the counterfactual images based on the following four criteria, each rated on a scale from 1 to 10:*

- *Validity: Does the counterfactual image successfully transition to the target attribute (i.e., male or female) in a visually convincing way?*

- *Sparsity: Are irrelevant attributes and regions left unchanged, with only the necessary parts being modified?*

- *Proximity: From a human perceptual perspective, how close is the counterfactual image to the original one?*

- *Realism: Does the counterfactual image remain close to the data manifold and distribution, i.e., does it appear as a realistic image that could exist in the original dataset?*

**BDD-OIA** *You are given two sets of animal images: one set contains the original images, and the other contains the corresponding counterfactual images. The target attribute is traffic status. If the traffic participants and infrastructure in the image indicate that the current traffic condition is stop or slow down, then the target is moving forward. Conversely, if the image reflects a moving forward traffic state, the target is to change it to stop or slow down. Please evaluate the counterfactual images based on the following four criteria, each rated on a scale from 1 to 10:*

- *Validity: Does the counterfactual image successfully transition to the target attribute (i.e., stop or moving forward) in a visually convincing way?*

- *Sparsity: Are irrelevant attributes and regions left unchanged, with only the necessary parts being modified?*

Sorrel to Zebra    Cougar to Cheetah    Persian Cat to Egyptian Cat

Input

CE

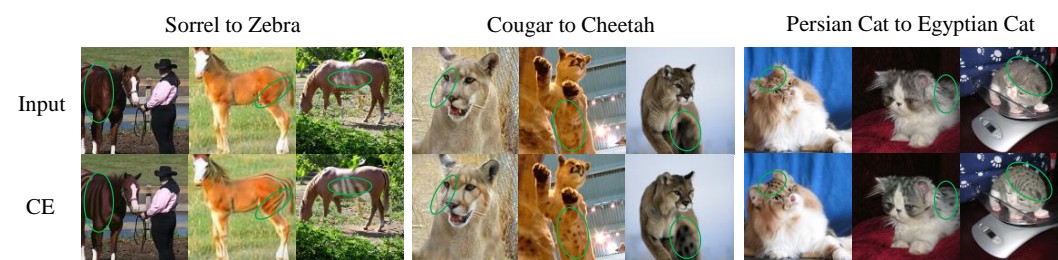

Figure 10: Extended qualitative results on ImageNet. Green circles highlight the counterfactual changes.

- *Proximity: From a human perceptual perspective, how close is the counterfactual image to the original one?*
- *Realism: Does the counterfactual image remain close to the data manifold and distribution, i.e., does it appear as a realistic image that could exist in the original dataset?*

**Sorrel - Zebra** *You are given two sets of animal images: one set contains the original images, and the other contains the corresponding counterfactual images. The target attribute to be modified is animal type. The query attribute is "sorrel", and the target attribute is "zebra". Please evaluate the counterfactual images based on the following four criteria, each rated on a scale from 1 to 10:*

- *Validity: Does the counterfactual image successfully transition to the target attribute (i.e., zebra) in a visually convincing way?*
- *Sparsity: Are irrelevant attributes and regions left unchanged, with only the necessary parts being modified?*
- *Proximity: From a human perceptual perspective, how close is the counterfactual image to the original one?*
- *Realism: Does the counterfactual image remain close to the data manifold and distribution, i.e., does it appear as a realistic image that could exist in the original dataset?*

**Cougar - Cheetah** *You are given two sets of animal images: one set contains the original images, and the other contains the corresponding counterfactual images. The target attribute to be modified is animal type. The query attribute is "cougar", and the target attribute is "cheetah". Please evaluate the counterfactual images based on the following four criteria, each rated on a scale from 1 to 10:*

- *Validity: Does the counterfactual image successfully transition to the target attribute (i.e., cheetah) in a visually convincing way?*
- *Sparsity: Are irrelevant attributes and regions left unchanged, with only the necessary parts being modified?*
- *Proximity: From a human perceptual perspective, how close is the counterfactual image to the original one?*
- *Realism: Does the counterfactual image remain close to the data manifold and distribution, i.e., does it appear as a realistic image that could exist in the original dataset?*

**Persian Cat - Egyptian Cat** *You are given two sets of animal images: one set contains the original images, and the other contains the corresponding counterfactual images. The target attribute to be modified is animal type. The query attribute is "Persian cat", and the target attribute is "Egyptian cat". Please evaluate the counterfactual images based on the following four criteria, each rated on a scale from 1 to 10:*

- *Validity: Does the counterfactual image successfully transition to the target attribute (i.e., Egyptian cat) in a visually convincing way?*

- *Sparsity: Are irrelevant attributes and regions left unchanged, with only the necessary parts being modified?*

- *Proximity: From a human perceptual perspective, how close is the counterfactual image to the original one?*

- *Realism: Does the counterfactual image remain close to the data manifold and distribution, i.e., does it appear as a realistic image that could exist in the original dataset?*

# D  ADDITIONAL PARAMETERS STUDY

In this section, we provide the parameter study plots for all datasets other than those in Section 5.5, where the observed patterns are consistent with those discussed in the main text. This highlights the critical role of $\alpha$ in our method, as it controls the degree of causal constraint on the change of spurious factors.

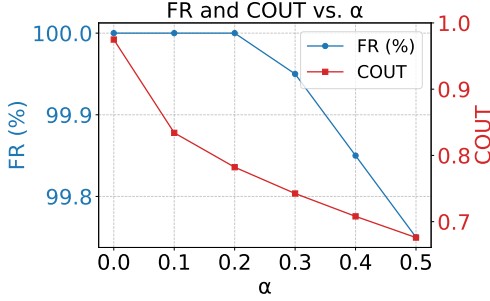

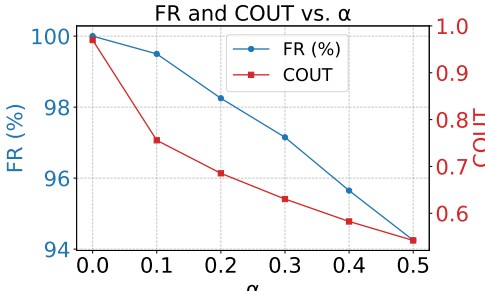

Figure 11: Line plots of FR and COUT with respect to varying values of $\alpha$ on CelebA Age.

Figure 12: Line plots of FR and COUT with respect to varying values of $\alpha$ on CelebA Gender.

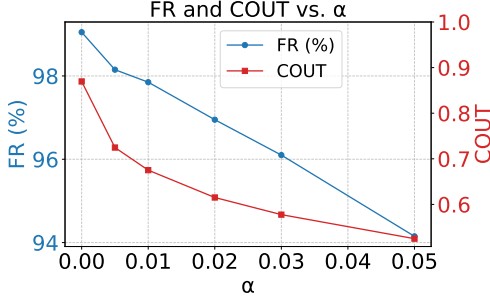

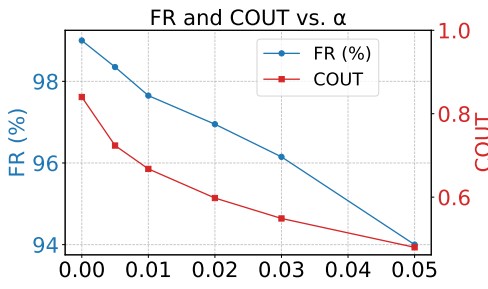

Figure 13: Line plots of FR and COUT with respect to varying values of $\alpha$ on CelebA-HQ Smile.

Figure 14: Line plots of FR and COUT with respect to varying values of $\alpha$ on CelebA-HQ Age.

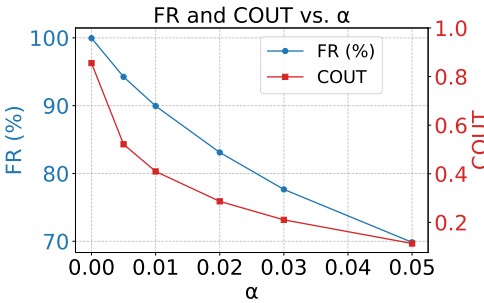

Figure 15: Line plots of FR and COUT with respect to varying values of $\alpha$ on CelebA-HQ Gender.

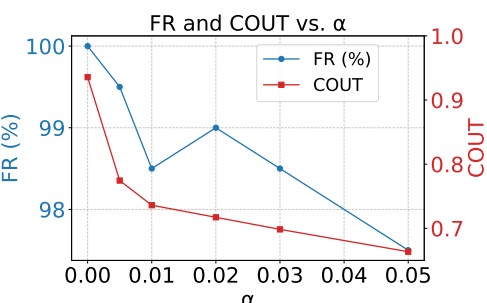

Figure 16: Line plots of FR and COUT with respect to varying values of $\alpha$ on BDD-OIA.

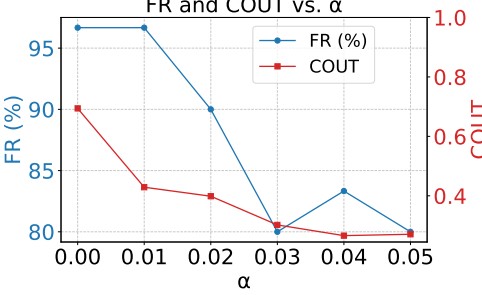

Figure 17: Line plots of FR and COUT with respect to varying values of $\alpha$ on ImageNet (Egyptian Cat - Persian Cat).

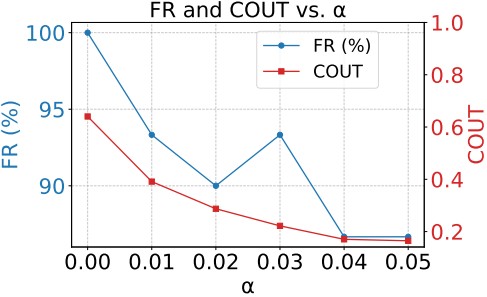

Figure 18: Line plots of FR and COUT with respect to varying values of $\alpha$ on ImageNet (Sorrel - Zebra).

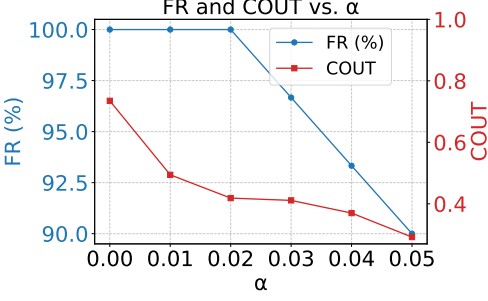

Figure 19: Line plots of FR and COUT with respect to varying values of $\alpha$ on ImageNet (Cougar - Cheetah).

