# OpenReview forum: "Counterfactual Visual Explanation via Causally-Guided Adversarial Steering"
_ICLR.cc/2026/Conference — Submitted to ICLR 2026_

### Official Review · Reviewer_Cpn8 · 2025-10-28

**Soundness:** 1
**Presentation:** 2
**Contribution:** 2
**Rating:** 2
**Confidence:** 4

**Summary:**

The paper proposes CECAS, a framework for generating causally guided counterfactual visual explanations. The method combines (1) a PGD-based adversarial perturbation guided by a causal penalty that aims to modify only causal (not spurious) features, and (2) a post-refinement stage that uses thresholding and diffusion-based inpainting to improve image quality. The authors claim this approach achieves better balance between validity, sparsity, proximity, and realism. Experiments on CelebA, CelebA-HQ, and several ImageNet subsets show improved quantitative metrics compared to prior diffusion-based counterfactual methods (ACE, DiME). They also evaluate realism with a vision-language model (GPT-4o) acting as a human-like judge.

**Strengths:**

- Ambitious goal of integrating causal reasoning with counterfactual visual explanation.
- Combines diffusion models with adversarial perturbations, which is technically non-trivial.
- Proposes a post-hoc refinement step that masks only important regions for modification.
- Includes experiments on multiple datasets with qualitative and quantitative results

**Weaknesses:**

1. **Weak motivation.** The central premise—that counterfactuals should change only causal, not spurious, features—is not convincingly justified. In explainability, highlighting spurious dependencies is *often desirable* to audit models. The paper gives no scenario where suppressing spurious cues yields better insight.
2. **Unclear and unsound causal mechanism.** The role of the auxiliary classifier \(g\) and the VAE-based decomposition \(R\!\to\!(C,S)\) is vague. The method trains \(g\) on “spurious” features to predict \(Y\), then penalizes the change in its logits, but this contradicts the notion of “spurious.” The causal factorization is neither theoretically grounded nor empirically validated.
3. **Missing ablations.** The architecture is complex but not decomposed experimentally. There is no evidence separating contributions from (a) the causal projection, (b) diffusion denoising, and (c) post-refinement. Qualitative examples suggest most gains come from the simple mask + inpainting step.
4. **Weak evaluation.**
   - Only a few datasets and two baselines are compared on all metrics.
   - Standard deviations are omitted (“negligible”), preventing assessment of significance.
   - No metrics measure *causal faithfulness* despite the causal framing.
   - Reliance on GPT-4o scoring lacks validation against human judgments.
5. **Poor figure and writing quality.** Figure 2 is overloaded and incomprehensible; several notations (orthogonalization, R′, g) are undefined in text.
6. **Omission of relevant literature.** Prior work on causal generative models and causal counterfactual generation (e.g., causal diffusion, SCM-based image editing) is ignored, overstating novelty.

**Questions:**

1. Why should we disregard spurious features in counterfactual explanations, given that exposing reliance on them is a key purpose of XAI? Provide a concrete use case where your approach yields more valuable insight.
2. Precisely how is the VAE trained to obtain \(C\) and \(S\)? What losses, architectures, and supervision are used? How do you verify that the split corresponds to causal versus spurious factors?
3. What is the exact role of the classifier $g$?
4. Provide ablations isolating each component: causal loss only, post-refinement only, full method. How much improvement stems from the simple inpainting step?
5. Discuss or cite causal counterfactual generation literature and clarify how your method differs.

---

> ### Author Response · Authors · 2025-11-28
>
> We thank the reviewer for the questions.
> - **W1:** We respectfully disagree with the reviewer’s point. One of the central goals of our work is to provide more precise explanations of the classifier’s behavior. **We aim to identify and explain the most essential causal factors that determine the classifier’s prediction.** Admittedly, spurious factors can also influence the classifier’s prediction. However, the counterfactual images generated by our method **aim to reveal, through truly minimal modifications, the most essential factors the classifier relies on.** For example, when generating a counterfactual image labeled as “old,” we expect the modifications to focus on facial wrinkles. A beard, on the other hand, should not be a necessary change, as elderly women typically do not grow beard.
> - **W2:** Our true objective in training g is not to use $S$ to predict $Y$, but rather to **leverage $Y$ as an anchor.** This allows us to align the logits obtained from $g$ when applied to the spurious factors $S$ of the original and counterfactual images, thereby constraining the distributional shift of spurious factors. The causal factorization is theoretically grounded in [1].
> - **W3:** We respectfully disagree with the reviewer’s view. The improvement in our qualitative results does not come from the mask and inpainting components, **as these modules are also present in the baseline methods, making the comparison fair.** We believe the most critical ablation is the removal of our causal loss, which constitutes our core contribution, and **we present this result in Section 5.5.** As for the diffusion denoising, it cannot be removed because our method relies on PGD attacks, which would fail to produce semantic-level changes without the diffusion-based denoising process.
> - **W4:**
>     - The datasets and baselines we use follows the settings of other existing baselines and are consistent with them. We plan to include additional ImageNet subsets in future versions to further expand the dataset.
>     - Due to the long runtime of the experiments, we conducted only three repeated trials. We observed that the standard deviations were on the order of one percent, which is why we chose to omit them.
>     - The focus of our paper is not on causal disentanglement itself, but on leveraging it to generate better counterfactual images. Therefore, it is reasonable and fair to evaluate our method using metrics designed for assessing counterfactual image quality.
>     - Thank you for point it out. However, due to the large scale of the counterfactual image set and our limited annotation resources, we were unable to include it. We will gradually incorporate human-in-the-loop judgments in the future
> - **W5:** We thank the reviewer for the suggestion, and we will update the corresponding image captions with more detailed descriptions.
> - **W6:** **Causal diffusion and SCM-based image editing approaches are fundamentally different from our work.** The core objective of our method is to **explain the behavior of a specific classifier, which requires that the generation of counterfactual images be explicitly dependent on that classifier.** While we do incorporate diffusion, **we do not utilize its generative capacity in the conventional sense.** Instead, the diffusion model and the classifier together form a **new robust classifier,** rather than serving as an image generator. In contrast, **causal diffusion is a purely generative model** and does not aim to explain any classifier. Similarly, **SCM-based image editing methods, such as CausalVAE, operate independently of any classifier**, they focus on modeling the data-generating process but are not designed for classifier interpretability.
> - **Q1:** See **W1**
> - **Q2:** For VAE training, we perform a do-operation on $S$, **we maximize the mutual information $I(C, Y | do(S))$ to ensure the effectiveness of the disentanglement.** The loss consists of two components: mutual information and KL divergence. The former is used to maximize the information flow from the causal factor $C$ to the label $Y$, while the latter ensures consistency between the data distributions before and after reconstruction. There are no specific structural requirements,both purely linear-layer VAEs and convolutional VAEs work well. We rely on the theoretical guarantees provided in [1] to ensure the effectiveness of the disentanglement.

---

> > ### Author Response · Authors · 2025-11-28
> >
> > - **Q3:** See **W2**
> > - **Q4:** If we completely remove the causal loss, the implementation becomes close to ACE, which we have already included in the comparison tables in our paper. If the goal is to observe the effect of varying the strength of the causal loss, we have provided corresponding results in Section 5.5. As for removing the post-refinement module, doing so would make the comparison unfair, since the baseline methods also employ the same technique.
> > - **Q5:** See **W6**
> >
> > [1] O'Shaughnessy, Matthew, et al. "Generative causal explanations of black-box classifiers." Advances in neural information processing systems 33 (2020): 5453-5467.

---

### Official Review · Reviewer_k1Gf · 2025-10-29

**Soundness:** 2
**Presentation:** 2
**Contribution:** 2
**Rating:** 2
**Confidence:** 3

**Summary:**

This paper proposes a new framework for producing images for counterfactual explanation. This framework, CECAS, integrates causal reasoning with adversarial perturbation methods. The primary contribution lies in addressing spurious correlations during counterfactual generation by constraining modifications to causally relevant features while preserving spurious factors unchanged. The approach combines adversarial attacks (PGD), diffusion models (DDPM), and a causal disentanglement mechanism to produce counterfactuals that achieve high validity, sparsity, proximity, and realism across multiple benchmark datasets.

**Strengths:**

- Strong motivation: The paper clearly articulates a significant weakness in existing visual counterfactual methods: they often latch onto spurious correlations (e.g., beards and age) rather than causal features, leading to unrealistic and unhelpful explainations.
- Introducing a causal perspective to guide the adversarial generation process is a novel and meaningful contribution. It directly tackles the reasoning behind poor counterfactuals.
- The experimental design tests the method across four distinct datasets with varying attributes (CelebA, CelebA-HQ, BDD-OIA, and ImageNet) and uses a robust set of quantitative metrics. So it's thorough.
- The authors incorporate a novel, human-like evaluation using Vision-Language Models. This adds a valuable layer of semantic validation to their claims.
- Convincing qualitative results: The visual examples demonstrate that CECAS avoids the unrealistic artifacts produced by other methods, such as the spurious beard growth on female faces during an age transition.
- The parameter study on the causal constraint weight (`α`) effectively demonstrates its impact on the trade-off between flip rate and causal faithfulness, reinforcing the method's core claims.

**Weaknesses:**

- Causal assumption is invalid (or at the least, unvalidated). The paper assumes the independence of causal (`C`) and spurious (`S`) factors (`C⊥S`)  but never validates it. This v-structure in Figure 1 is a strong assumption.
- The VAE-based disentanglement is underspecified, which is a major weakness . The paper provides no details on the VAE architecture, its training, or any quantitative or qualitative evidence that the disentanglement was successful. Without this, there is no proof that the `Ls` loss is actually constraining spurious factors; it's just constraining an arbitrary, unvalidated latent subspace.
- The baselines are narrowly focused on other recent generative methods (STEEX, DiVE, DiME, ACE). A comparison against a wider range of non-diffusion CE methods would have strengthened the paper.
- The VLM evaluation is presented without any scientific rigor. There is no discussion of prompt bias, nor reliability (e.g., running the same prompt multiple times), and no validation against human-in-the-loop judgments.
- In the mask generation equation, the paper divides the pixel difference by M, which it defines as the maximum value after the summation. Is M calculated once per image, or is it a constant for all images, or calculated based on a local window? M is ambiguous, and hence not reasonably reproducible.
- In equation 9, if `m=0`, $ [(1-m)x_t + mx_t^0] $ becomes $ x_t $. If `m=1`, it becomes $  x_t^0 $. This tells the model to keep the counterfactual in the unchanged areas and the original image in the changed areas? This seems backwards (or I could be misunderstanding it).
-  In Table 2, the metrics contradict the claims of the paper. Correlation Difference (CD) score for CECAS is 5.33, worse than the baseline DiME (4.00). The authors don't explain satisfactorily why CD should be discounted.

**Questions:**

Restating or repeating some of the weaknesses:
1.  In Table 2, the metrics contradict the claims of the paper. Correlation Difference (CD) score for CECAS is 5.33, worse than the baseline DiME (4.00). The authors don't explain satisfactorily why CD should be discounted. Could you elaborate on why your causally guided method performed worse on this specific metric?
2. Can you expand on the inpainting logic? it's possible that the $ x_t $ and $  x_t^0 $ are flipped in the equation.
3. The paper's central claim rests on the VAE's ability to disentangle C and S. Are there any quantitative or qualitative evidence that this disentanglement was successful and how it was validated?

---

> ### Author Response · Authors · 2025-11-28
>
> We thank the reviewer for the questions.
> - **W1:** We acknowledge that this assumption is strong, but **we believe it is reasonable.** Accurately modeling the full data-generating process with a causal graph is inherently difficult and not the primary focus of our work. Instead, **we leverage external human knowledge to construct a simplified causal graph.** Specifically, we assume that an image is composed of two types of factors: causal factor C, also referred to as the content variable, which directly and exclusively determines the classifier’s prediction; and spurious factor S, also referred to as the style variable, which is unrelated to the prediction. Therefore, C and S are assumed to be independent. Several prior works have also adopted the same assumption [1][2][3].
> - **W2:** Our main focus and contribution lie in **introducing a causal perspective for explaining classifier behavior through counterfactual images.** The VAE is merely a tool we adopt to perform causal disentanglement. In the original work [4] that proposed this VAE, **the model achieved strong performance and was supported by detailed theoretical analysis.** Nevertheless, **we will include the specific VAE architecture and training details** in the appendix of a future version.
> - **W3:** These baselines have **similar pipelines and closely related objectives to ours,** which is why we selected these four models for comparison. Other non-diffusion-based methods may differ structurally, but when applied to classifier interpretability, they share the same fundamental goal. We respectfully ask whether the reviewer could suggest a few representative non-diffusion models, we would be happy to attempt comparisons with them.
> - **W4:**
>     - **Prompt design:** We carefully constructed the prompts based on the four standard dimensions for evaluating counterfactual images (validity, sparsity, proximity, and realism), as shown in the appendix. These prompts are designed to be concise and neutral, aiming to minimize bias as much as possible.
>     - **Reliability:** We ran the evaluation three times using the same prompt and averaged the results to reduce the effect of randomness in the VLM outputs.
>     - **Human-in-the-loop** This is a great suggestion. However, due to the large scale of the counterfactual image set and our limited annotation resources, we were unable to include it. We respectfully ask for the reviewer’s understanding. We will gradually incorporate human-in-the-loop judgments in the future.
> - **W5:** **$M$ is calculated based on a local $d\times{d}$ window.** In Equation (8), the summation is computed only over pixels at coordinates $(u, v)$, where **$(u, v)$ belongs to the $d \times d$ local neighborhood around pixel $(i, j)$ (as indicated in the annotation beneath the max operator).**
> - **W6:** We thank the reviewer for pointing it out. It was caused by a typo in our Equation (8). **The “>” sign in Equation (8) should be replaced with “<”.** With this correction, **the modified region corresponds to a mask value of 0 while the unmodified region corresponds to 1,** which explains why the mask needs to be inverted before applying it to the counterfactual image in Equation (9).
> - **W7:** First, underperforming on a single metric does not imply that our method is overall inferior to the baseline. Second, **a higher CD value does not necessarily indicate incorrect modifications to causal factors.** For example, editing the attribute “age” may naturally influence around five other attributes on average. Since the CelebA dataset **does not provide a ground truth causal graph,** the CD metric has certain limitations, **when the values are close, it is not reliable to judge performance solely based on their numerical differences.**
> - **Q1:** See **W7**
> - **Q2:** See **W6**. $x_t$ and $x_t^0$ are not flipped. Typo is in equation (8).
> - **Q3:** See **W2**. We use a modified version of the VAE compared to the one in the original work. Specifically, we perform disentanglement on the output features $f$ of the classifier’s feature extractor, rather than directly on the raw images. This makes qualitative analysis more difficult. Moreover, since our goal is not generalization but targeted causal control, we train the VAE by maximizing $I(C, Y \mid do(S))$, which ensures that the learned representation $C$ captures the causal factors that determine the prediction $Y$.

---

> > ### Author Response · Authors · 2025-11-28
> >
> > [1] Kotovenko, Dmytro, et al. "Content and style disentanglement for artistic style transfer." Proceedings of the IEEE/CVF international conference on computer vision. 2019.
> >
> > [2] Ren, Xuanchi, et al. "Rethinking content and style: exploring bias for unsupervised disentanglement." Proceedings of the IEEE/CVF International Conference on Computer Vision. 2021.
> >
> > [3] Zhang, Yonggang, et al. "Causaladv: Adversarial robustness through the lens of causality." arXiv preprint arXiv:2106.06196 (2021).
> >
> > [4] O'Shaughnessy, Matthew, et al. "Generative causal explanations of black-box classifiers." Advances in neural information processing systems 33 (2020): 5453-5467.

---

### Official Review · Reviewer_CoB5 · 2025-10-30

**Soundness:** 3
**Presentation:** 2
**Contribution:** 2
**Rating:** 4
**Confidence:** 3

**Summary:**

The paper introduces a method for counterfactual image generation called CECAS. The core idea is to take a causal perspective on the image generation process to avoid spurious features, and instead focus only on the causally relevant ones. The paper's experiments show their method achieves better scores on their metrics of counterfactual flip rate and other plausibility metrics.

**Strengths:**

The paper addresses an interesting problem of causality in images, and appears to avoid some unnecessary changes in the images based on the figures.

The experimental evaluation is heavily in their method's favor, suggesting it is a worthwhile contribution.

An extra qualitative evaluation adds weight to the actual image quality.

**Weaknesses:**

In general, my main critique of the paper is that it appears to be quite incremental in its contributions. There are quite a few papers addressing the issue of counterfactual image generation, and the contribution of this one is not overly clear to me.

For example, in the paper's claimed contributions they state "We provide a new perspective on the problem of counterfactual visual explanation by highlighting the critical role of causality". But please see e.g. [1], this just isn't true at all, plenty of work has highlighted this.

Most of the evaluation only compares against a single method ACE, which is quite weak.

I also wonder of the validity of these UNet-based CF methods (e.g. [2]) in light of current image generation tools such as Nano Banana, which can generate far superior image edits to these approaches. I understand of course that adapting such an approach to this would be non-trivial given the whole setup, but far more interesting in my opinion (and likely far more useful).

I am quite wary of metrics such as FID, I would much rather the authors evaluate the usefulness of the method compared to baselines in downstream tasks. That would be better and more convincing.

***

[1] Melistas, T., Spyrou, N., Gkouti, N., Sanchez, P., Vlontzos, A., Panagakis, Y., Papanastasiou, G. and Tsaftaris, S.A., 2024. Benchmarking counterfactual image generation. Advances in Neural Information Processing Systems, 37, pp.133207-133230.

[2] He, Z., Zuo, W., Kan, M., Shan, S. and Chen, X., 2019. Attgan: Facial attribute editing by only changing what you want. IEEE transactions on image processing, 28(11), pp.5464-5478.

**Questions:**

Can you please explain how this makes a significant contribution beyond the myriad of CF image generation methods? You seem to claim you are the first to consider the separation of causal v. non-causal features, and that is the core contribution. This seems a bit of a stretch to me given work such as (but hardly limited to) these [1, 2].

Do you have any indication your method would be useful for tasks such as model improvement, HCI collaboration, teaching humans concepts learned by the model etc...

***

[1] Kocaoglu, M., Snyder, C., Dimakis, A.G. and Vishwanath, S., 2018, February. CausalGAN: Learning Causal Implicit Generative Models with Adversarial Training. In International Conference on Learning Representations.

[2] Pawlowski, N., Coelho de Castro, D. and Glocker, B., 2020. Deep structural causal models for tractable counterfactual inference. Advances in neural information processing systems, 33, pp.857-869.

---

> ### Author Response · Authors · 2025-11-28
>
> We sincerely thank the reviewer for the questions and fully understand the concern behind them.
> - **W1: Contribution is not clear.** We apologize for the lack of clarity in our previous statement. Unlike [1], which **primarily relies on VAE and GAN to generate counterfactual images and uses the classifier only for evaluation,** our work focuses more on using counterfactual images to **explain the classifier,** with the images themselves also being generated through the gradient of classifier. Therefore, the two works differ in their core emphasis.
> - **W2: Most of the evaluation only compares against a single method ACE.** On the ImageNet dataset, we indeed compared against only one baseline, ACE. This is because the other baselines either did not report results on this dataset or could not be reproduced due to unavailable or non-functional code. Among the models that could be successfully run, ACE is the best-performing one. Moreover, based on the results observed on other datasets, it is reasonable to expect that the other baselines would not outperform ACE, and therefore would also not outperform our method.
> - **W3: The validity of these UNet-based CF methods.** Many generative models are **built upon diffusion models,** whose backbone is typically a U-Net architecture. Nano Banana is also likely based on this type of architecture. While these powerful models are indeed capable of producing high-quality edited images, **such edits are often difficult to use for explaining classifier behavior,** and it becomes challenging to ensure that the modifications are truly minimal.
> - **W4:Wary of metrics such as FID.** At the current stage, **FID is indeed a strong evaluation metric, as it reflects the realism of the generated counterfactual images,** which in turn indicates the extent and scope of the modifications. As for evaluating usefulness through downstream tasks, we agree this is a highly promising direction, and we plan to continue exploring it in future work.
> - **Q1:** Our method provides a causal perspective for explaining classifier predictions. Specifically, by modifying causal factors to generate counterfactual images, we aim to answer the question: **What is the truly minimal change needed to alter the classifier's prediction?** [2] does incorporate causal mechanisms, but it **relies on GANs** and does not provide interpretability for the classifier. [3] learns a structured causal model composed of invertible deep mechanisms. It first infers the exogenous noise from the observed image, then performs interventions on the causal variables and regenerates the image using the same noise to obtain the counterfactual. **This process does not rely on a classifier and similarly does not focus on interpretability.**
> - **Q2:** **Model improvement:** By revealing the minimal causal changes required to flip a classifier’s prediction, our method helps identify which features the model relies on. This can guide targeted data augmentation or regularization strategies to mitigate spurious correlations and improve robustness. **HCI collaboration:** Our causal counterfactual images can serve as a transparent interface for communicating model behavior to users, enabling more intuitive interactions. For example, users can visually understand what a model considers essential for a decision, fostering trust and aiding in model debugging. **Teaching humans model-learned concepts:** The counterfactual edits produced by our method highlight what the model has implicitly learned as discriminative features. This can help domain experts or lay users understand abstract concepts encoded in the model in a visual, concrete way.
>
> [1] Melistas, T., Spyrou, N., Gkouti, N., Sanchez, P., Vlontzos, A., Panagakis, Y., Papanastasiou, G. and Tsaftaris, S.A., 2024. Benchmarking counterfactual image generation. Advances in Neural Information Processing Systems, 37, pp.133207-133230.
>
> [2] Kocaoglu, M., Snyder, C., Dimakis, A.G. and Vishwanath, S., 2018, February. CausalGAN: Learning Causal Implicit Generative Models with Adversarial Training. In International Conference on Learning Representations.
>
> [3] Pawlowski, N., Coelho de Castro, D. and Glocker, B., 2020. Deep structural causal models for tractable counterfactual inference. Advances in neural information processing systems, 33, pp.857-869.

---

### Official Review · Reviewer_vRh9 · 2025-10-30

**Soundness:** 1
**Presentation:** 1
**Contribution:** 1
**Rating:** 2
**Confidence:** 4

**Summary:**

This paper considers the problem of counterfactual explanation (CE) generation for classifiers in computer vision. The proposed approach adapts the ACE method to additionally limit the editing of spuriously correlated factors. This is achieved by adding a variational autoencoder (VAE) trained on the representations of the explained model, which are later fed into an auxiliary classifier trained to mimic the behavior of the original one. The resulting misalignment is used to regularize the optimization procedure. Empirically, the method is evaluated on a broad range of datasets and problems.

**Strengths:**

S1. The proposed approach is evaluated on a representative set of datasets with metrics covering the entire evaluation spectrum present in the current literature on CEs.

**Weaknesses:**

W1. The proposed methodology is flawed at its core. The authors propose generating CEs with an additional constraint on the spurious correlations learned by the explained model that limits their appearance in the explanations. However, the primary role of CEs is to actually reveal these correlations, since they highlight the model's unreasonable failure cases, which is desirable.

W2. The paper claims to be outperforming *recent state-of-the-art* methods for CE generation, while ignoring the actual recent papers on this topic that have appeared in top conferences in the last two years. For example, the paper is missing references to OCTET [1], DiG-IN [2] and RCSB [3]. These baselines are crucial, as they consider similar techniques for constraining the changes in the CEs. Moreover, the paper properly cites DVCE [4], which is a work concurrent to DiME and ACE, but fails to compare with it in any way.

W3. The novelty of the method is limited and the authors do not properly credit the ACE paper [5]. A quick glance at the paper's implementation shows that it is entirely based on ACE's codebase. The method is presented in a way that frames both phases (adversarial-attack-based and refinement) as contributions of the paper, while both are entirely taken from ACE. The only remaining novelty is the addition of the factor-disentangling VAE and the auxiliary classifier, a contribution that is logically flawed (W1).

W4. The paper does not provide any further details on the training procedure for the VAE and the auxiliary classifier, which seem to be the primary novelty claimed by the authors. Does the VAE require causal and spurious concept annotations? If yes, is the method limited exclusively to datasets that provide them? If not, how does its accuracy influence the effectiveness of CECAS? What is the error of the auxiliary classifier? These are just a few sample questions that are never addressed in this work.

W5. As the authors correctly point out, adversarial attacks (AA) result in imperceptible changes that influence the model's decision. Incorporating AA techniques into CE generation risks obtaining a modified image where the semantic edits are not actually responsible for the decision change, but rather the small AA-based pixel changes. The same can be said about the ACE method.

W6. Many less significant errors, some of which are pointed out below, show that the work is currently unpolished and requires crucial refinement:

1.  line 058: The actual problem with standard diffusion-based models is that they do not operate in semantically meaningful latent spaces, but rather in high-dimensional pixel spaces.
2.  "Two-colored nodes" are nowhere to be seen in Figure 1.
3.  Table 2 incorrectly bolds and emphasizes the results. It only considers the authors' method and not others.
4.  The Figure 4 caption incorrectly references the line plot of FR vs COUT. Why do the results of ACE and the proposed CECAS not differ perceptually? Is it because these methods are essentially the same on ImageNet (W3)?
5.  The $\alpha$ variable is introduced in an ambiguous way. It first appears as the step size for the PGD attack (Eq. 5), then is used as the regularization strength for the second term in Eq. 6. Moreover, figures like Figure 4 once again highlight its flawed influence - it restricts the range of edits to deviate from the spuriously correlated ones.
6.  lines 051-053: This example is inherently wrong, since the shape of a traffic sign is not a spurious factor, but rather a property used to characterize types of signs. Stating that either content or shape is *causally relevant to the model’s decision* is rather far-fetched, as such a statement requires access to the ground-truth of the model's explanation, which is almost always unavailable.

[1] Zemni et al., OCTET: Object-aware Counterfactual Explanations, CVPR, 2023

[2] Augustin et al., DiG-IN: Diffusion Guidance for Investigating Networks - Uncovering Classifier Differences Neuron Visualisations and Visual Counterfactual Explanations, CVPR, 2024

[3] Sobieski et al., Rethinking Visual Counterfactual Explanations Through Region Constraint, ICLR, 2025

[4] Augustin et al., Diffusion Visual Counterfactual Explanations, NeurIPS, 2022

[5] Jeanneret et al., Adversarial Counterfactual Visual Explanations, CVPR, 2023

**Questions:**

Please refer to the weaknesses above.

---

> ### Author Response · Authors · 2025-11-28
>
> We sincerely thank the reviewer for the valuable suggestions.
> - **W1:** We respectfully disagree with the reviewer’s point. One of the central goals of our work is to **provide more precise explanations of the classifier’s behavior.** It is well known that, without any constraints, classifiers tend to rely on spurious factors for prediction. Counterfactual images are intended to reveal the minimal changes needed to flip a classifier’s prediction. **If spurious factors are involved in this process, the resulting explanations become ambiguous.** For example, consider the task of minimally modifying an image to flip the classifier’s prediction from “young” to “old.” Without proper constraints, some counterfactual images of women might include the addition of beard. However, beard does not accurately explain the classifier’s decision regarding age, as elderly women typically do not grow beard. **Our work aims to uncover the fundamental causal factors driving the classifier’s predictions, rather than providing vague or entangled explanations that mix causal and spurious cues.**
> - **W2:** The reason we did not compare with [1], [2], and [4] is that **none of these works focus on minimizing the extent of changes.** As can be seen from their results, **the modifications in their counterfactual images are often significant.** Moreover, **none of them report FID scores,** which makes quantitative comparison difficult. In particular, **[2] leverages Stable Diffusion and allows image manipulation via prompts, making a direct comparison unfair** due to the substantial difference in generative capabilities. **This can significantly affect the interpretability of the model’s behavior.** Moreover, **the code of [3] are not reproducible (data zip file downloaded from the official code repository consistently failed to decompress properly),** which is why we did not include it in our comparison. Nonetheless, **it is still a good work, and we will make sure to cite it appropriately in the related work section.**
> - **W3:** We would like to clarify that **we did not claim the introduction of adversarial attacks as our contribution;** in fact, we explicitly **provided a citation in line 224.** Our main contributions begin from line 235. Furthermore, **as respectfully stated in W1, we do not agree that there is a logical flaw in our stated contributions.**
> - **W4:**
>     - Our work **primarily aims to provide a new causal perspective** for explaining classifiers through counterfactual images. The **training procedures of other modules are not the main focus.** However, we will consider **including additional details** on the training of the VAE and the auxiliary classifier in future versions.
>     - We adopt the technique from [6] to train the VAE, which **does not require additional concept annotations**, only the original classification labels are needed. The VAE training process is robust, as we are not concerned with generalization performance. Achieving over 95% accuracy is straightforward, so **the impact of VAE training accuracy on CECAS is negligible.**
>     - The auxiliary classifier also does not require high accuracy. During the counterfactual image generation process, we only need the logits ẑ and ẑ′ to be as similar as possible (as stated in line 249). The specific values of these logits are not important, **our goal is to align the spurious distribution before and after the modification.**
> - **W5:** We would like to clarify that for standard, non-robust models, adversarial attacks do generate imperceptible changes to influence model decisions. However, **when targeting robust classifiers, adversarial attacks are forced to produce semantic-level changes**, which has been demonstrated in [7].

---

> > ### Author Response · Authors · 2025-11-28
> >
> > - **W6:**
> >     - In line 58, we are simply summarizing the original description from [8], where the intermediate noised states of DDPM are referred to as the latent space.
> >     - We apologize for the incorrect image and will correct it promptly.
> >     - We thank the reviewer for pointing this out and will make the necessary correction.
> >     - The caption of Figure 4 is correct. **This figure is intended to illustrate parameter study,** and therefore **it only shows how our method's performance** on the FR and COUT metrics varies with different values of $\alpha$.
> >     - We thank the reviewer for pointing out the typo. We will replace the symbol $\alpha$ used in PGD with a different notation.
> >     - We respectfully disagree with the reviewer’s view to some extent. While it is true that a model may rely on both content and shape to predict a traffic sign, **this does not mean that all factors used for prediction are causally relevant.** A straightforward example is that a model might rely on background information to classify animals, even though the background is clearly not a causal factor. In our traffic sign example, we aim to illustrate that **if the shape of a stop sign changes, the model should ideally maintain its prediction, because the sign still conveys the meaning of “stop”.** However, **if the content of the sign changes, the model should alter its prediction**, for instance, classifying it as a speed limit sign. Therefore, in the context of predicting the function of a traffic sign, we consider content as the causal factor and shape as the spurious factor.
> >
> > [1] Zemni et al., OCTET: Object-aware Counterfactual Explanations, CVPR, 2023
> >
> > [2] Augustin et al., DiG-IN: Diffusion Guidance for Investigating Networks - Uncovering Classifier Differences Neuron Visualisations and Visual Counterfactual Explanations, CVPR, 2024
> >
> > [3] Sobieski et al., Rethinking Visual Counterfactual Explanations Through Region Constraint, ICLR, 2025
> >
> > [4] Augustin et al., Diffusion Visual Counterfactual Explanations, NeurIPS, 2022
> >
> > [5] Jeanneret et al., Adversarial Counterfactual Visual Explanations, CVPR, 2023
> >
> > [6] O'Shaughnessy, Matthew, et al. "Generative causal explanations of black-box classifiers." Advances in neural information processing systems 33 (2020): 5453-5467.
> >
> > [7] Pérez, Juan C., et al. "Enhancing adversarial robustness via test-time transformation ensembling." Proceedings of the IEEE/CVF International Conference on Computer Vision. 2021.
> >
> > [8] Jeanneret, Guillaume, Loïc Simon, and Frédéric Jurie. "Diffusion models for counterfactual explanations." Proceedings of the Asian conference on computer vision. 2022.

---

### Official Review · Reviewer_b5zk · 2025-11-01

**Soundness:** 2
**Presentation:** 3
**Contribution:** 2
**Rating:** 4
**Confidence:** 3

**Summary:**

This paper proposes CECAS (Counterfactual Explanation via Causally Adversarial Steering), a framework for generating counterfactual visual explanations. The method combines an adversarial perturbation stage with a causal regularization term​, which aims to preserve spurious (non-causal) features while modifying causal ones, using a VAE-based disentanglement of causal and spurious representations. A second stage refines the resulting images using ℓ₁-based masks and diffusion-based inpainting to improve realism and sparsity. Experiments on CelebA, CelebA-HQ, BDD-OIA, and ImageNet subsets show that CECAS achieves higher flip rates and better perceptual quality metrics (FID, LPIPS) than existing counterfactual explanation methods such as DiVE, STEEX, DiME, and ACE.

**Strengths:**

- The paper tackles an important problem in counterfactual visual explanation by aiming to make generated examples causally faithful and semantically meaningful.

- The experimental evaluation is extensive and covers multiple datasets, metrics, and strong baselines, which supports the empirical effectiveness of the results.

- The paper is clearly written and easy to follow, with well-structured sections and illustrative figures that make the proposed method understandable.

**Weaknesses:**

My main concern with this paper is that the proposed method is overly complex and lacks conceptual elegance. The pipeline combines several loosely connected components: a PGD-based adversarial attack, a VAE for causal–spurious disentanglement, a mask extraction step, and diffusion-based inpainting. Each of them introduces its own set of hyperparameters and loss terms. While these parts collectively improve visual quality, the overall design feels more like a layered engineering solution than a coherent causal formulation. The causal guidance is implemented through an additional regularization term rather than a principled causal mechanism, and the diffusion refinement serves mainly as a post-hoc fix to clean artifacts instead of addressing their root cause within the generation process. As a result, the framework appears heavy, ad-hoc, and difficult to interpret or reproduce, which weakens the overall conceptual clarity and novelty of the work.
- The causal guidance mechanism is largely heuristic and depends on a VAE-based disentanglement that is not guaranteed to separate true causal and spurious factors.
- The method still operates in pixel space, so adversarial perturbations may flip predictions through imperceptible or non-semantic changes rather than meaningful causal edits.
- There is limited evidence that the generated counterfactuals truly preserve spurious features or modify causal ones, as no causal validation or ablation analysis is provided.
- The diffusion refinement improves realism but can also re-generate image regions, which weakens the interpretability of the counterfactual as a minimal intervention on the same instance.
- The GPT-4o–based vision-language evaluation is interesting but may introduce bias and lacks reproducibility or robustness analysis.
- Some qualitative figures are cluttered with red and green circles that distract from the visual comparison and are not essential to interpret the results.
- In Figure 5, the counterfactual edits for gender conversion appear too subtle to convincingly demonstrate a change in the intended attribute. The “Male to Female” counterfactuals still largely resemble male faces, and the “Female to Male” results also retain feminine features. This suggests that the method may struggle to produce sufficiently strong or semantically distinct attribute changes for certain categories.
- The quality and reliability of the generated counterfactuals heavily depend on the classifier used to guide them. If the classifier itself relies on spurious correlations or non-causal features, the resulting counterfactuals will likely reflect and reinforce those same biases rather than reveal true causal changes. This raises concerns about whether the method can produce meaningful explanations when the underlying model is imperfect or biased.
- The overall pipeline involves multiple ad-hoc steps, including PGD-based adversarial attack, VAE-based disentanglement, mask computation, and diffusion-based inpainting, which makes the approach complex and less principled.
- The diffusion-based refinement serves mainly as a post-hoc fix to remove artifacts, rather than addressing the underlying cause of imperfect or unrealistic counterfactual generation within the main optimization process.

**Questions:**

- How well does the proposed causal–spurious disentanglement generalize across datasets, and how sensitive are the results to the choice of the VAE architecture or latent dimensionality?
- Have you quantitatively verified that the causal loss truly preserves spurious features while altering causal ones?
- Since the method depends on the classifier’s gradients, how robust are the counterfactuals if the classifier itself relies on spurious correlations?
- Can the authors provide evidence that the generated counterfactuals remain perceptually and semantically distinct when evaluated by an independent or robust classifier?
- How consistent are the GPT-4o evaluation results across different prompt formulations or repeated runs?
- Could the causal guidance be implemented more elegantly, for example, directly in latent space or via a unified generative causal model, rather than through multiple post-hoc components?
- Does the causal guidance term risk over-constraining the perturbation, leading to weaker or insufficient edits as seen in the Male↔Female examples?

---

> ### Author Response · Authors · 2025-11-28
>
> We sincerely appreciate the reviewer’s feedback and constructive questions.
> - **W1:** We build upon the VAE-based method for disentangling causal and spurious factors proposed in [1]. A reasonable assumption is that the causal factor $C$ and the spurious factor $S$ are independent. Then, by performing a do-operation on $S$, **we maximize the mutual information $I(C, Y | do(S))$ to ensure the effectiveness of the disentanglement.**
> - **W2:** We would like to clarify an important point: For standard, vulnerable models, adversarial attacks can indeed flip the classifier’s prediction by introducing imperceptible perturbations. However, **when targeting robust classifiers, adversarial attacks are forced to produce larger, semantically meaningful changes[2].** In our work, we adopt such a robust classifier, namely, a diffusion-based joint classifier. The filtering preprocessing provided by the diffusion model enhances robustness without modifying the original classifier’s weights. **Combined with our causally guided module, this enables the generation of meaningful causal edits to the image.**
> - **W3:** The focus and main contribution of our work lie in **leveraging causal mechanisms to improve the quality of counterfactual generation and enhance the interpretability of the classifier.** The core of our work **does not lie in causal disentanglement.** In prior counterfactual generation works, the generated counterfactual images often involve changes to spurious factors, for example, in generating a counterfactual image labeled as “old”, some females may be shown with beard. This leads to poor performance in terms of validity, sparsity, proximity, and realism. Our work precisely **leverages these four metrics to indirectly reflect the effectiveness of the causal module, which is well aligned with our motivation and contributions.**
> - **W4:** During training, we use an $L1$ loss to obtain a binary mask that segments the main modified region in the image. This region is then fused with the corresponding unmodified part from the original image, which leads to a visible blending boundary. To remove this boundary and make the image appear more realistic, we adopt a diffusion-based refinement technique. However, **we apply noise to the entire fused image uniformly,** without relying on diffusion to generate new content in areas where nothing existed before, and **the noise level is not strong enough to resemble pure noise.** As a result, **the overall distribution of the image remains stable.** Furthermore, the refinement module is introduced to enable a fair comparison with one of the baselines, and **ablating this module results in only a negligible impact on the overall experimental performance.**
> - **W5:** We adopt GPT-4o as an evaluation tool to leverage the rich prior knowledge of large models for a more comprehensive assessment of counterfactual image quality, going beyond traditional metrics. During evaluation, we use the **average results over multiple runs,** ensuring that the experimental results are reproducible and reasonably robust.
> - **W6:** We thank the reviewer for the helpful suggestion. Our original intention in adding markers was to help readers more easily locate the subtle differences between the counterfactual images and the original ones. To address this concern, we will consider removing the circles and instead providing a separate column with zoomed-in image comparisons in future revisions.
> - **W7:** Our goal is to generate counterfactual images that help explain what specific changes are necessary to flip the classifier’s prediction. At the same time, counterfactual images are expected to satisfy validity, sparsity, proximity, and realism. This implies that **stronger semantic changes are not always better, as such changes may significantly compromise the interpretability of the model.** Moreover, our method **relies solely on classification labels as supervision,** which inherently provide limited semantic guidance. Unlike other counterfactual generation approaches that utilize more powerful models such as Stable Diffusion along with carefully crafted prompts for image synthesis, **our work places greater emphasis on classifier interpretability.** Therefore, it is reasonable that the modifications in our generated counterfactual images are subtle.
> - **W8:** We would like to clarify that **our work is precisely aimed at addressing the concern raised.** Many classifiers do not explicitly disentangle causal and spurious factors during training, which may lead them to rely on spurious cues for prediction. This is **one of the key motivations** behind our work. **We perform disentanglement in the latent space of the classifier and introduce a loss that aligns spurious factors, thereby providing causal guidance for counterfactual image generation.**

---

> > ### Author Response · Authors · 2025-11-28
> >
> > - **W9:** We thank the reviewer for the suggestion. However, **these components are essential** parts of our method. **Both the PGD and the VAE-based disentanglement are theoretically grounded.** The mask computation and diffusion refinement are designed to further enhance the quality of the generated images. Our contribution lies in integrating these components in a way that allows each to **serve its intended purpose effectively and to its fullest potential.**
> > - **W10:** As we stated in W4, the diffusion refinement module is indeed designed to eliminate artifacts. However, the underlying cause of imperfect or unrealistic counterfactual generation **is addressed within the main pipeline** through causal disentanglement and alignment loss. **These two components are not contradictory but rather complementary.**
> > - **Q1:** First, as shown in the tables in both the main text and the appendix, our proposed method **demonstrates effectiveness across multiple datasets.** Otherwise, it would not be possible for the counterfactual images generated by our method to perform well in terms of validity, sparsity, proximity, and realism. In addition, prior to the main experiments, we conducted extensive preliminary testing, including experimenting with VAE architectures composed purely of linear layers as well as those based on CNNs. We also tested various latent space dimensions (32, 64, 256, 512). **We found that the VAE-based disentanglement is highly robust, and increasing the architectural complexity or latent dimensionality yields only marginal improvements.**
> > - **Q2:** As we stated in W3, the focus of our work is not on proving the effectiveness of causal disentanglement itself. Nonetheless, our experimental results indirectly demonstrate the effectiveness of this module: **our generated images achieve a higher prediction shift rate (COUT) with lower FID scores.** Moreover, the effectiveness of VAE-based causal disentanglement **has already been validated in [1].**
> > - **Q3:** As we mentioned in W8, **our work is specifically aimed at addressing this issue.** Many classifiers rely on spurious factors for prediction. To tackle this, **we first disentangle the causal and spurious factors,** and then apply an **additional loss to constrain variations caused by the spurious factors.** This guides the optimization **toward a gradient direction that is more aligned with the true causal direction.**
> > - **Q4:** **We have already provided relevant metrics in the paper, namely FR and COUT.** The former directly measures the proportion of counterfactual images that successfully flip the classifier’s prediction, while the latter measures the transition scores between the original and counterfactual images. Since our work is centered around generating counterfactual images based on a specific classifier and explaining its predictions, it is appropriate to **use the same classifier for evaluation.**
> > - **Q5:** We **carefully designed the prompts** based on the four dimensions used to evaluate counterfactual images, as shown in the appendix. We did not design multiple sets of prompts because the **most precise and effective prompts are often limited to just a few or even a single formulation.** Under multiple runs with the same prompt, **our method consistently outperforms all other methods.**
> > - **Q6:** We thank the reviewer for the forward-looking suggestion, this is indeed a very promising direction. However, at the current stage, implementing such an approach remains challenging. Our method offers a practical and feasible solution.
> > - **Q7:** **Causal guidance does not lead to over-constraining.** First, we provide the flip rate (FR) in our quantitative analysis, if the model were over-constrained, the FR would be low, possibly even lower than the baseline. However, **our method achieves the highest FR.** Second, as shown in the qualitative analysis, our approach effectively **prevents the generation of spurious perturbations and reduces meaningless artifacts such as random color patches.**
> >
> >
> > [1] O'Shaughnessy, Matthew, et al. "Generative causal explanations of black-box classifiers." Advances in neural information processing systems 33 (2020): 5453-5467.
> >
> > [2] Pérez, Juan C., et al. "Enhancing adversarial robustness via test-time transformation ensembling." Proceedings of the IEEE/CVF International Conference on Computer Vision. 2021.

---

### Meta-Review · Area_Chair_yrRU · 2026-01-06

**Summary:**

The primary concerns center on the validity of the core causal assumptions and the completeness of the evaluation. Reviewers (vRh9, k1Gf, Cpn8) strongly challenged the unvalidated assumption of independence between causal and spurious factors and the underspecified VAE-based disentanglement mechanism.  Additionally, there was a consensus that the paper omits critical recent baselines (e.g., DiG-IN, OCTET) and relies too heavily on a single comparison (ACE), with reviewers rejecting the authors' claim that generative-based methods are incomparable. Finally, the "ad-hoc" complexity of combining PGD, VAEs, and diffusion refinement without sufficient ablations led to skepticism regarding the method's novelty and rigorousness compared to existing work.

**Reviewer Concerns:**

While the authors successfully clarified technical details regarding equation typos and the mask generation logic, the most substantive concerns remain unaddressed. The authors' refusal to benchmark against broader state-of-the-art methods—citing architectural differences rather than functional utility—failed to satisfy reviewers who view counterfactual generation as a unified task. Furthermore, the fundamental disagreement regarding the goal of explainability remains unresolved; reviewers maintain that suppressing spurious correlations obscures model flaws, while the authors argue it reveals "true" decision boundaries, a philosophical deadlock that the rebuttal did not bridge. Finally, the reliance on prior theory rather than empirical evidence to validate the specific VAE disentanglement in this context left the causal claims unsubstantiated.

**Reviewer Scores:**

It is highly unlikely that the dissenting reviewers (vRh9, k1Gf, Cpn8) would raise their scores, as their fundamental critiques regarding the "flawed" causal methodology and missing baselines were met with deflection rather than new empirical evidence. Reviewer vRh9 would almost certainly maintain a rejection given the disagreement on the utility of spurious features, while Reviewer k1Gf would remain unconvinced by the promise of future appendix details regarding the VAE. Even the borderline reviewers (b5zk, CoB5) would likely stagnate or lower their scores, as the rebuttal reinforced the perception of the method as an incremental, complex engineering solution rather than a principled breakthrough.

---

### Decision · Program_Chairs · 2026-01-26

Reject